# Structural basis of SALM5-induced PTPδ dimerization for synaptic differentiation

Zhaohan Lin[1], Jianmei Liu[1], Huandi Ding[1], Fei Xu[1] & Heli Liu[1]

SALM5, a synaptic adhesion molecule implicated in autism, induces presynaptic differentiation through binding to the LAR family receptor protein tyrosine phosphatases (LAR-RPTPs) that have been highlighted as presynaptic hubs for synapse formation. The mechanisms underlying SALM5/LAR-RPTP interaction remain unsolved. Here we report crystal structures of human SALM5 LRR-Ig alone and in complex with human PTPδ Ig1–3 (MeA⁻). Distinct from other LAR-RPTP ligands, SALM5 mainly exists as a dimer with LRR domains from two protomers packed in an antiparallel fashion. In the 2:2 heterotetrameric SALM5/PTPδ complex, a SALM5 dimer bridges two separate PTPδ molecules. Structure-guided mutations and heterologous synapse formation assays demonstrate that dimerization of SALM5 is prerequisite for its functionality in inducing synaptic differentiation. This study presents a structural template for the SALM family and reveals a mechanism for how a synaptic adhesion molecule directly induces *cis*-dimerization of LAR-RPTPs into higher-order signaling assembly.

---

[1] State Key Laboratory of Natural and Biomimetic Drugs & Department of Molecular and Cellular Pharmacology, School of Pharmaceutical Sciences, Peking University Health Science Center, 38 Xueyuan Road, Haidian District, Beijing 100191 China. Correspondence and requests for materials should be addressed to H.L. (email: liuheli@hsc.pku.edu.cn)

The human brain consists of billions of neurons that exploit a myriad of proteins to establish synaptic connections in precise patterns for processing information, shaping neural circuits, and maintaining proper brain functions[1, 2]. Synaptic adhesion molecules (SAMs) are precisely positioned to sculpt synaptic protein interaction networks for mediating synaptic early differentiation, formation, maturation, terminal elimination, and plasticity[3, 4]. Disruption of synaptic protein networks involved by SAMs may contribute to the pathology of many neurological disorders[4, 5].

SALMs, also known as leucine-rich and fibronectin III domain-containing proteins, are a family of newly characterized SAMs that belong to the type I transmembrane proteins. The SALM family includes five members, SALM1–5, that share a similar domain organization: a leucine-rich repeat (LRR) domain composed of LRRs flanked by cysteine-rich regions at its amino terminus (LRRNT) and carboxy terminus (LRRCT), an Ig-like domain, and a fibronectin type III (Fn) domain, a transmembrane region and a cytoplasmic tail that binds to PSD-95 for SALM1–3, but not for SALM4–5[5–7]. Members of the SALM family are predominantly expressed in the central nervous system (CNS) and known for their engagement in neurite outgrowth, synapse formation, and synapse maturation[7–15]. All five SALMs may form heteromeric and homomeric complexes in cis- or trans- configurations, suggesting that SALMs exhibit both pre- and post-synaptic functions through protein interactions[16]. Notably, SALM3 and SALM5 may induce both excitatory and inhibitory presynaptic differentiation in contacting axons via trans-synaptic interactions with LAR-RPTPs[12, 13, 15]. Interestingly, SALM4 may suppress excitatory synapse development by cis-inhibiting trans-synaptic SALM3-LAR adhesion[14]. SALM5 has recently been associated with familial schizophrenia in which inherited copy number variation of SALM5 has been observed[17], and with severe progressive autism, where expression levels of SALM5 are dramatically reduced by tenfold due to chromosomal translocation and intragenic multiple-exon deletion[18, 19]. These reports not only point to the clinical importance of SALMs, but also suggest that SALM5-dependent presynaptic induction may play critical roles in brain development and function[7, 15].

LAR-RPTPs, short for leukocyte common antigen-related receptor protein tyrosine phosphatases, are classified as type IIa RPTPs that, in vertebrates, include three members, LAR, PTPδ, and PTPσ. LAR-RPTPs are type I transmembrane proteins and share a common domain architecture of an ectodomain containing up to three immunoglobulin-like (Ig) domains and up to nine fibronectin III (Fn) repeats, followed by a transmembrane domain and tandem intracellular phosphatase domains. Multiple LAR-RPTP isoforms are generated by alternative splicing of whole domains or four short peptides encoding mini exon inserts, termed MeA to MeD[20–22]. Distinct from the other subtype of RPTPs, LAR-RPTPs interact trans-synaptically with a growing list of post-SAMs including NGL-3[23, 24], TrkC[25], IL1RAPL1[26, 27], IL-1RAcP[28], the Slitrks[29, 30], and the newly found SALM family members, SALM3[13] and SALM5[15]. LAR-RPTPs may also bind to presynaptic molecules, netrin-G1[31] and glypicans[32], in a cis manner. In particular, the MeA and MeB insertion sites within the N-terminal Ig domain of the LAR-RPTPs have received extensive attention since they are involved in regulating interactions with various post-synaptic ligands, like TrkC, IL-1RAcP, IL1RAPL1, and SALM5[15, 22]. Through interacting with such a large collection of protein ligands, LAR-RPTPs exhibit a wide repertoire of cellular signaling functions and have recently been highlighted as hubs for extracellular interactions in neurons, regulating neurite outgrowth, axon guidance, synaptic organization, and CNS regeneration[22, 33, 34]. In humans, PTPδ has been implicated in correlation with neurological disorders such as ASDs, ADHD, bipolar disorder, schizophrenia, and restless legs syndrome[15, 35–39].

Early structural analysis has focused on elucidating catalytic mechanism and substrate specificity of the intracellular phosphatase domains of RPTPs[40], whereas the architectures and functions of the extracellular regions lack complete understanding[22]. Recently, structural work on the ectodomains of LAR-RPTPs has demonstrated their surprising flexibility as a valuable feature that allows these receptors to serve as a synaptic signaling nexus for interacting with a broad spectrum of different protein ligands in the narrow synaptic cleft[22, 41]. Also, structural analyses of type IIa RPTPs in complexes with Slitrk[42], TrkC[41], IL1RAPL1, or IL-1RAcP[43] and Slitrk2[44] have revealed the molecular basis for their interaction specificity as well as minimal units and 1:1 stoichiometry for receptor–ligand recognition. However, how extracellular ligands mediate receptor dimerization or clustering for downstream signaling has remained outstanding questions in the RPTP field[45].

In contrast to other known ligands of the type IIa RPTPs, SALM5 forms a dimer in solution[46], which could confer itself a unique mechanism for RPTP signaling. Therefore, deciphering the codes for SALM5 dimerization and its interaction with LAR-RPTPs at the structural level will yield novel insights into the molecular mechanisms underlying RPTP transmembrane signal transduction. Here we report crystal structures of human SALM5 LRR-Ig alone and in complex with human PTPδ Ig1–3 (MeA−). Together with other biophysical, biochemical, and cellular evidence, the structures reveal that dimerization of SALM5 is prerequisite for its functionality in inducing synaptic differentiation through presynaptic LAR-RPTP receptors. Also, this study provides a structural template for the SALM family and shed light on how a SAM directly induces oligomerization of type IIa RPTPs into higher-order signaling assembly.

## Results

**Overall structure of the 2:2 SALM5/PTPδ heterotetramer.** In order to solve the SALM5 and SALM5/PTPδ complex structures, we screened expression and purification of human SALM5 LRR-Ig or its full-length ectodomain alone and in complex with human PTPδ proteins. Since the Ig domains of LAR are sufficient to mediate SALM5 binding[15], we focused on the N-terminal three Ig domains of human PTPδ with or without MeA or MeB splice inserts (Fig. 1a). All the constructs were expressed successfully using the baculovirus-transduced HEK293S cells and purified into high purity and homogeneity. After many unsuccessful crystallization trials for glycan-trimmed SALM5 or SALM5/PTPδ complexes, we finally gained diffraction quality crystals of human SALM5 LRR-Ig alone and in complex with PTPδ Ig1–3 (MeA−). The crystal structure of human SALM5 LRR-Ig domains was solved with an iodine derivative using the single isomorphous replacement and anomalous scattering (SIRAS) method, while the structure of SALM5/PTPδ complex was solved using the molecular replacement method (Table 1).

For the SALM5/PTPδ complex crystal, the asymmetric unit contained one dimeric complex, in a 2:2 stoichiometry, where a central SALM5 dimer bridges two monomeric PTPδ molecules together. The complex has three dimensions of ~120 Å × 100 Å × 75 Å. Overall, the shape of the heterotetrameric complex resembles the Simuwu Rectangle Ding, an ancient Chinese bronze ware, where two SALM5 LRR domains constitute the Ding body and the SALM5 Ig domains, together with the Ig3 domains of two PTPδ molecules, act as four feet to buttress the body (Fig. 1b, c). Although tethered by the SALM5 dimer, each PTPδ Ig1–3 molecule interacts with only one SALM5 protomer, and no interaction between two PTPδ Ig1–3 molecules is observed. The C termini of two PTPδ Ig3 domains are 98 Å

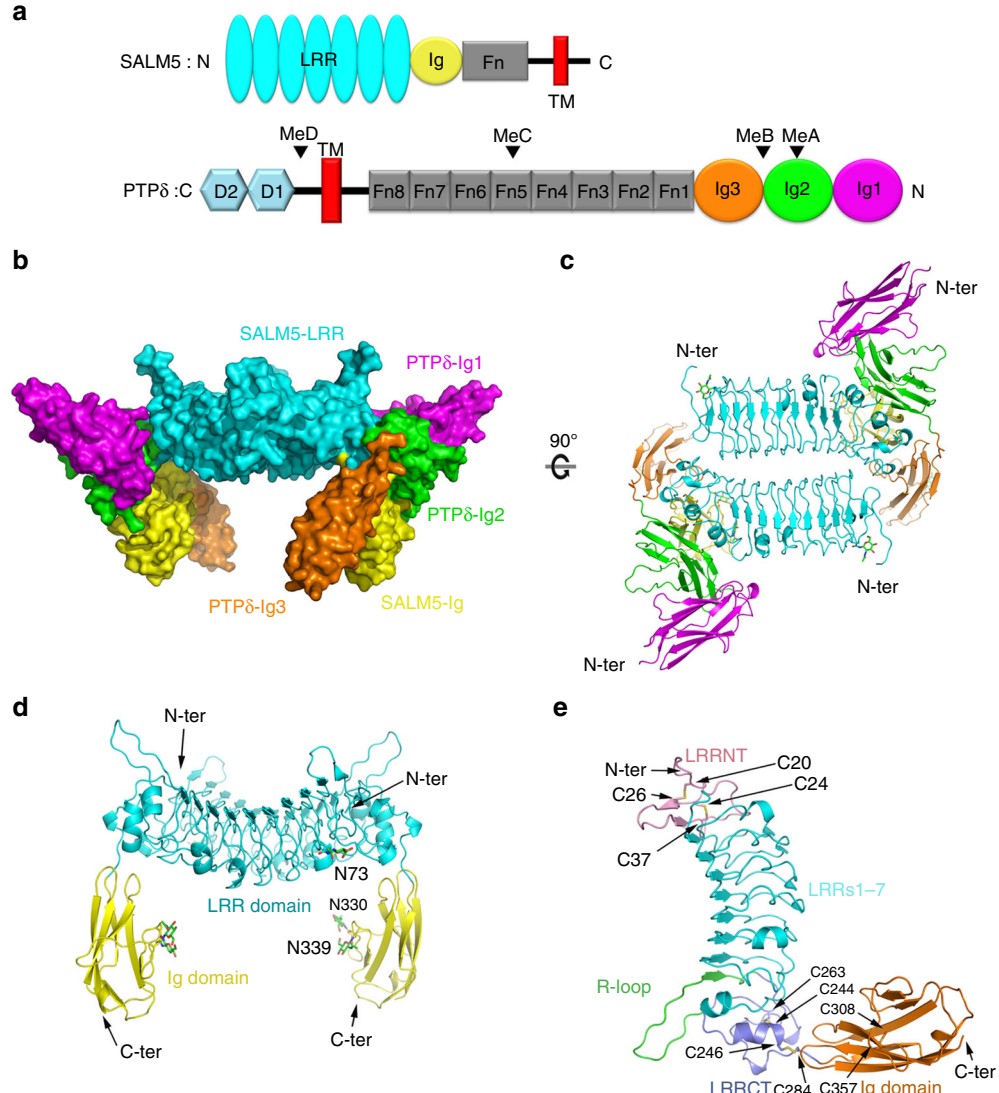

**Fig. 1** Overall structures of the 2:2 heterotetrameric SALM5/PTPδ complex, SALM5 dimer, and monomer. **a** Domain organization of human SALM5 and human PTPδ. LRR: leucine-rich repeats domain; Ig: Ig-like domain; Fn: fibronectin type III domain; D1, D2: a tandem of tyrosine–protein phosphatase domains; Me: mini exon; TM: transmembrane domain; N: N-terminus; C: C-terminus. **b** Surface representation of the dimeric SALM5/PTPδ complex. The LRR domain and Ig domain in SALM5 are colored cyan and yellow, respectively. The Ig1, Ig2, and Ig3 domains in PTPδ are colored magenta, green, and orange, respectively. **c** Top view of the dimeric SALM5/PTPδ complex shown in ribbons. **d** Side view of the SALM5 dimer (extracted from PTPδ-bound complex structure) in ribbons. **e** Ribbon representation of SALM5 LRR-Ig domains. Disulfide bonds are indicated using yellow sticks. Color codes are as follows: LRRNT, pink; LRRs 1–7, cyan; LRRNT, slate; R-loop, green; Ig domain, orange. **c**, **d** The N-linked glycans are indicated as sticks, and three glycosylation sites (N73, N330, and N339) of one SALM5 molecule are labeled. The N and C terminus of each molecule are, respectively, labeled as "N-ter" and "C-ter"

apart and extend to an opposite direction. Given that the PTPδ ectodomain comprises a tandem of fibronectin domains arranged in a manner similar to its homolog PTPσ[41], the remaining Fn domains would emanate away from the central dyad axis and the extracellular region of two PTPδ molecules would not interact with each other in the signaling complex.

The asymmetric unit of the free SALM5 crystal contained one SALM5 molecule. Except two loops, most residues in the Ig domain of SALM5 were not determined in the free SALM5 crystal due to lack of density, suggesting dynamic movement or multiple conformations of the Ig domain (Supplementary Fig. 1a). As demonstrated by crystal packing analysis (Supplementary Fig. 1b–d) and structural superimposition (Supplementary Fig. 1e), the free and PTPδ-bound SALM5 structures unanimously revealed that two SALM5 molecules form a diad dimer through their LRR domains packing in an antiparallel side-by-

side manner. The LRR domains of two SALM5 molecules constitute a concave platform as the core of the dimer, with approximate dimensions of 56 Å × 68 Å. In each PTPδ-bound SALM5 protomer, the Ig domain keeps its long axis perpendicular to the platform; the two Ig domains in the PTPδ-bound SALM5 dimer are in parallel with each other and no contacts are observed between them (Fig. 1d). The C termini of the two PTPδ-bound SALM5 Ig domains are at a distance of 78 Å, supposedly ushering the Fn domain of SALM5 to a proper position on the cell membrane. The antiparallel side-by-side packing architecture of the dimer is likely in an ideal orientation for *cis*-interaction of two post-synaptic SALM5 molecules.

**Structure of SALM5 LRR-Ig domains**. The SALM5 LRR-Ig totally adopts a totally all-β fold, with the overall structure in an

**Table 1 Data collection, phasing, and refinement statistics (SIRAS)**

| | Native SALM5 for refinement | Native SALM5 for SIRAS | NaI-derivative of SALM5 for SIRAS | SALM5/PTPδ |
|---|---|---|---|---|
| *Data collection* | | | | |
| Space group | P6₃22 | P6₃22 | P6₃22 | I23 |
| Cell dimensions | | | | |
| $a$, $b$, $c$ (Å) | 152.601, 152.601, 93.822 | 152.752, 152.752, 93.63 | 152.801, 152.801, 93.32 | 249.384, 249.384, 249.384 |
| $\alpha$, $\beta$, $\gamma$ (°) | 90, 90, 120 | 90, 90, 120 | 90, 90, 120 | 90, 90, 90 |
| Resolution (Å) | 50–2.80 (2.85–2.80)[a] | 50–2.85 (2.9–2.85) | 50–3.66 (3.72–3.66) | 50–3.73 (3.79–3.73) |
| $R_{merge}$ | 5.6 (58.8) | 5.9 (51.8) | 16.3 (58.9) | 10.9 (91.9) |
| $I$ / $\sigma I$ | 36.9 (3.4) | 62.1 (7.7) | 48.0 (12.0) | 18.8 (2.5) |
| Completeness (%) | 98.8 (99.2) | 100 (100) | 100 (100) | 99.8 (99.9) |
| Redundancy | 6.4 (6.6) | 18.9 (20.1) | 41.2 (42.9) | 7.1 (7.2) |
| *Phasing* | | | | |
| Resolution range (Å) | | | 50–3.78 | |
| $R_{ano}$ (%) | | | 6.49 | |
| $R_{iso}$ (%) | | | 14.3 | |
| Figure of merit | | | 0.48 | |
| *Refinement* | | | | |
| Resolution (Å) | 2.80 | | | 3.73 |
| No. reflections | 16115 | | | 26924 |
| $R_{work}$/$R_{free}$ | 24.8/26.9 | | | 23.2/26.4 |
| No. atoms | | | | |
| Protein | 2238 | | | 10166 |
| Ligand/ion | 28 | | | 158 |
| Water | 156 | | | 94 |
| B-factors | | | | |
| Protein | 86.0 | | | 133.0 |
| Ligand/ion | 111.5 | | | 183.2 |
| Water | 82.2 | | | 104.7 |
| R.m.s. deviations | | | | |
| Bond lengths (Å) | 0.003 | | | 0.003 |
| Bond angles (°) | 0.805 | | | 0.739 |

[a] Values in parentheses are for highest-resolution shell

L-shape (Fig. 1e). The LRR domain resembles a solenoid with its concave side lined with ten long parallel β-strands and convex side comprising variable loops, short β-strands and 3₁₀ helical segments. The LRR domain consists of seven LRR repeats (LRR1–7), flanked by an N-terminal LRRNT (residues 14–49) and a C-terminal LRRCT (residues 240–285) subdomain. The LRRNT subdomain is mainly composed of an β-hairpin plus two loops cross-linked by two pairs of disulfide bonds C20–C26 and C24–C37. The LRRCT subdomain folds into loops interwoven with short helices that are bridged to loops by disulfide bonds C244–C263 and C246–C308. Interspaced between the LRR repeats and LRRCT is the tenth long β-strand plus a long loop, which is termed "R-loop" as in platelet glycoprotein Ibα[47]. Both LRRNT and LRRCT serve as a cap to shelter the LRR repeats from solvent environment through hydrophobic residue contacts (Supplementary Fig. 2a, b). Different from previous sequence alignment that suggested SALM5 contains six LRRs[7], we assumed that SALM5 has seven LRR repeats (Fig. 1e and Supplementary Fig. 2). The discrepancy mainly lies in how to define the first LRR repeat. Although T53 replaces the first leucine residue in the

consensus sequence motif, LxxLxLxxNxL, that defines a typical LRR repeat[48], we still assigned the segment (residues 50–73) to the first LRR repeat because (1) T53 projects its $C_\gamma$ atom to the hydrophobic core of this segment involved by residue L56; (2) this segment adopts a configuration similar to that of a canonical LRR repeat (Supplementary Fig. 2a). Indeed, in the case of glycoprotein Ibα, the first leucine residue in the consensus LRR motif is also substituted with a threonine residue[47]. Other structural features of LRR and Ig domains of SALM5 are further analyzed in Supplementary Fig. 2b–e.

**SALM5 dimerization mainly governed by hydrophilic residues.** Dimerization of SALM5 is exclusively mediated by its LRR domain. The two protomers in the SALM5 dimer are related by a twofold axis, and the dimer interface buries a total of ~2200 Å² solvent accessible surface area (SASA) with a long list of residues involved (Supplementary Table 2). Viewed along the twofold symmetry axis, the dimer interface can be divided into five sites with the site A flanked by other four sites, of which the sites B and C are symmetrically equivalent to B′ and C′, respectively (Fig. 2a). The sites C and C′ reside in both ends of the dimer. For clarity, only the interactions at the sites A, B, and C are described in detail as follows.

Both in the sites A and B, the residues engaged in dimerization are from the ascending loops connecting two adjacent β-sheets. In the site A, two sets of ascending loops in the LRRs 3–6 are intertwined at the dimer interface, and an extensive hydrogen network is built by engagement of residues R110, Q134, and N158. The guanidine group of the residue R110 from one protomer forms four hydrogen bonds with the side chain amide group of N158 and the main chain carbonyl oxygen atoms of N157 and H180 from the other protomer. In addition, Q134 and N158 from two protomers project their side chain amide groups to each other and form two hydrogen bonds (Fig. 2b). In the site C, the dimer interface is constituted by the loops in the LRRNT subdomain from one SALM5 molecule and those in the LRRCT from the other molecule. R23 extends its side chain to the interior of the dimer interface, and its guanidine group forms two hydrogen bonds with the main chain carbonyl of G271 and the hydroxyl group of T262. Also, the main chain carbonyl group of K40 and G41, respectively, form an hydrogen bond with the side chain hydroxyl group of Y273 and T262 (Fig. 2c). Flanking these four hydrogen bonds are two hydrophobic patches. One patch is on the convex surface of the dimer, consisting of L43 and F44 from one SALM5 molecule, as well as A264 and the methyl group of T270 from the other molecule. The other small patch is located on the concave surface of the dimer and consists of L260 and the aromatic group of Y273. The two layers of hydrophobic patches provide a low dielectric-constant environment that may enforce the hydrogen bonds engaged by the sandwiched hydrophilic residues. Located in the middle of the sites A and C, the site B includes a few van der Waals interactions and one hydrogen bond formed by residues F62, T64, and T86 from one SALM5 molecule and S204 and K206 from the other. Collectively, the SALM5 dimer interface is mainly mediated by hydrophilic residues, which form a network of hydrogen bonds, and interspersed with a few hydrophobic residues.

**Disrupting SALM5 dimerization may impair its functionality.** Since in line with our crystal structures, SALM5 has recently been reported to mainly exist in solution as a dimer[46], we dissected the functional role of SALM5 dimerization in mediating presynaptic differentiation. In order to obtain monomeric SALM5, we exploited a "glycan wedge" strategy of introducing glycosylation sites on the dimer interface. As demonstrated by mass

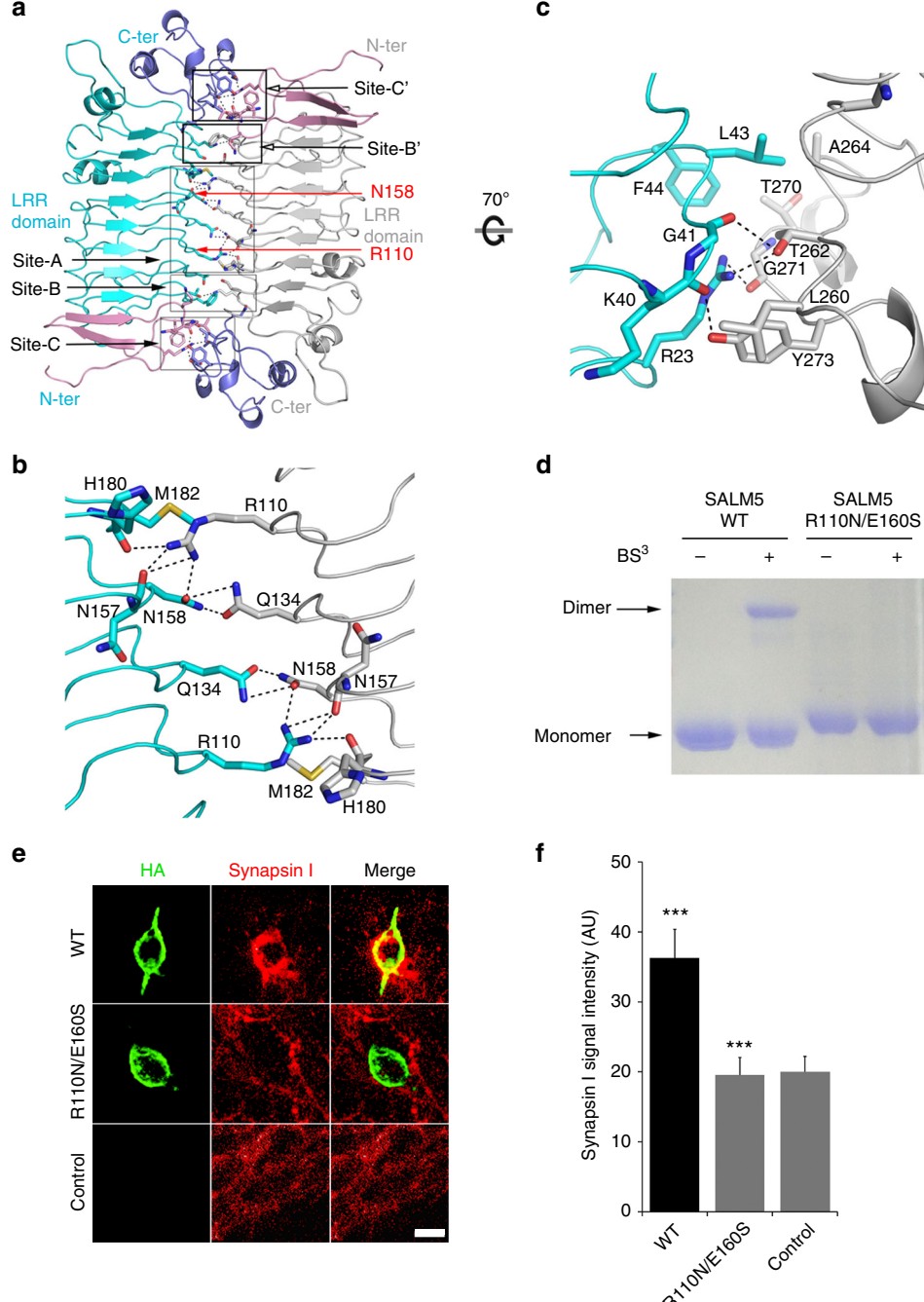

**Fig. 2** Structural, biochemical, and functional characterizations of SALM5 dimerization. **a** Top view of the dimeric SALM5 LRR domains (from the 2.8 Å free SALM5 structure) represented in ribbons with residues on the dimer interface displayed as sticks. The LRR domains of two protomers are colored in cyan and gray, respectively. For clarity, R110 and N158 from one protomer are labeled. The N and C terminus of each molecule are, respectively, labeled as "N-ter" and "C-ter". LRRNT: pink; LRRCT: slate; oxygen atoms: red; nitrogen atoms: blue; hydrogen bonds and salt bridges: black dashed lines. **b** Close-up view of the site A in **a**. **c** Close-up view of the site C in **a**, with a 70° rotation along the horizontal line. **d** SDS-PAGE analysis of the wild-type SALM5 LRR-Ig and its R110N/E160S mutant in the presence or absence of BS$^3$. **e** Representative images of the heterologous synapse formation activities of the wild-type full-length SALM5 ectodomain and its mutant R110N/E160S. Scale bar, 20 μm. The neurons were co-cultured with HEK293T cells transfected with plasmids for the HA-tagged wild-type SALM5 (WT) and its mutant (R110N/E160S) or empty vector (control), and stained with antibodies against HA-tag (green) and synapsin-I (red). The synapse-forming activity was quantified by measuring the intensity of immunostained synapsin-I around HEK293T cells. **f** Quantification of the result in **e**. $n = 21$ fields of view, ***$p < 0.005$ for WT vs control or mutant vs WT, Student's $t$ test. Error bars present standard error of the mean (S.E.M.)

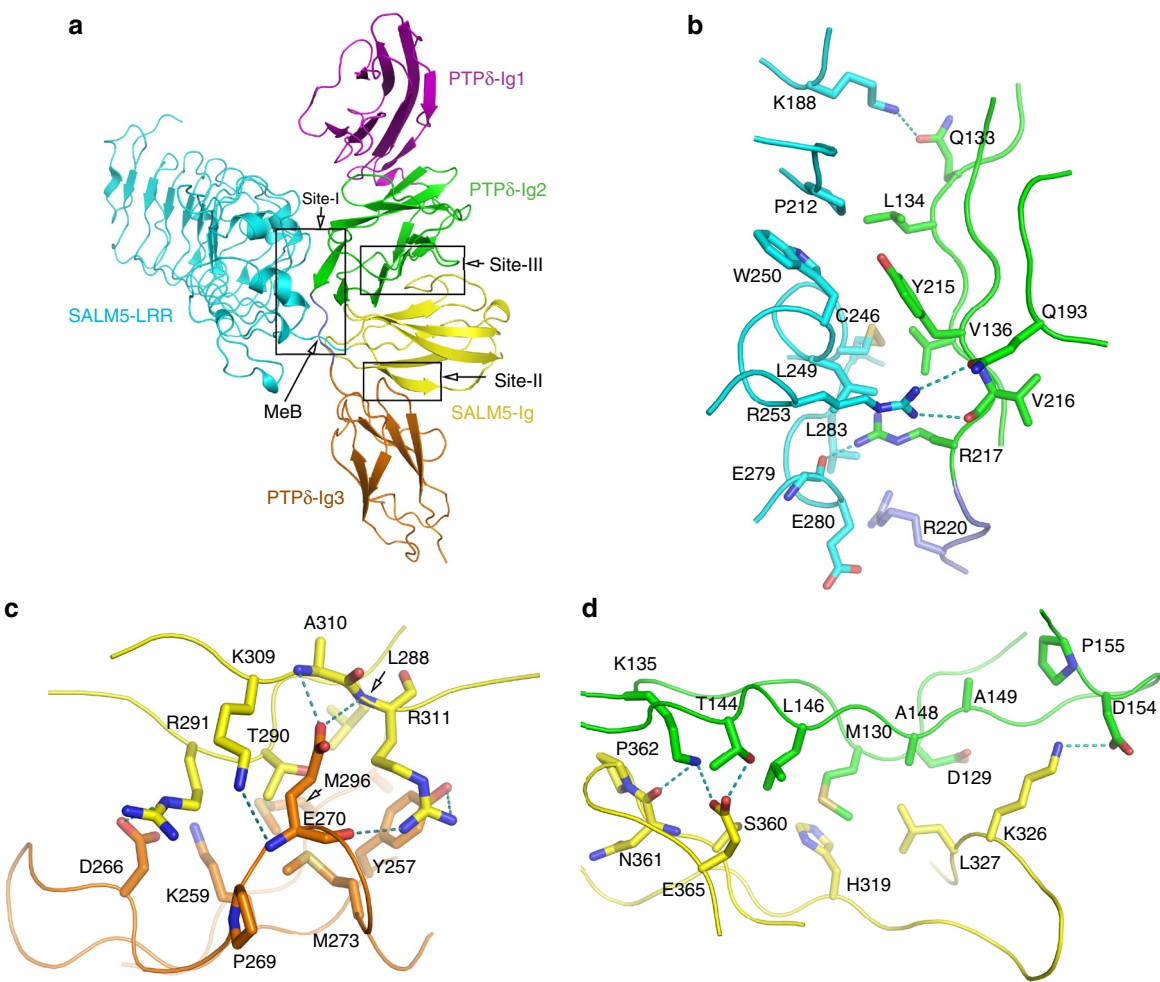

**Fig. 3** Recognition mechanism of SALM5 by PTPδ. **a** Ribbon representation of 1:1 SALM5/PTPδ complex, with the same color code for each domain as Fig. 1a. The complex interface is divided into three sites labeled as site I, II, and III. **b**–**d** Close-up views of site I (**b**), II (**c**), and III (**d**). **c** is relative to **a** by a 90° rotation along the vertical line. **a**, **b** The MeB insert and its residue R220 are colored in slate

spectrometry assay (Supplementary Fig. 3a), the double mutation of R110N and E160S led to addition of N-linked glycans at residues R110 and N158, both of which inhabit the center of the SALM5 dimer interface (Fig. 2a). Analytical ultracentrifugation (AUC) study indicated that the R110N/E160S mutant mainly exists as a monomer in solution because the mutant has an observed molecular mass of 42.1 kDa, close to a theoretical molecular mass of 44.8 kDa (Supplementary Fig. 4). As expected, the wild-type SALM5 LRR-Ig cross-linked by the chemical BS³ (*bis*[sulfosuccinimidyl] suberate) migrated as a dimer on a SDS-PAGE gel. In comparison, the R110N/E160S mutant moved as a monomer in the presence or absence of BS³, further indicating that the SALM5 R110N/E160S mutant is a monomer in solution (Fig. 2d). The monomeric SALM5 R110N/E160S mutant is still capable of binding to PTPδ and forms a stable complex on a gel-filtration column with an elution volume apparently larger than that for the wild-type SALM5/PTPδ complex (Supplementary Fig. 5). Because synapsin-I clustering is a hallmark for SALM5-dependent presynaptic differentiation[15], we compared intensities of immunostained synapsin-I clustering in primary hippocampal neurons co-cultured with HEK293T cells that displayed the wild-type full-length ectodomain of SALM5 and its variant R110N/E160S. In contrast with the HEK293T cells transfected with the empty vectors, the wild-type SALM5 displaying cells could trigger clustering of synapsin-I in the neurons with a statistically significant increase in the mean fluorescence

intensity of immunostained synapsin-I. However, the SALM5 mutant R110N/E160S cells did not increase synapsin-I clustering level as compared with the control group, instead significantly decreased in comparison with the wild-type SALM5 cells (Fig. 2e, f). This meant that R110N/E160S mutation abrogated the capacity of SALM5 to induce presynaptic differentiation. Taken together, disrupting SALM5 dimerization may significantly impair its functionality in mediating presynaptic differentiation. BS³ (*bis*[sulfosuccinimidyl] suberate)Please be noted that a serious error was introduced here during your production process. In our manuscript, BS³ represents a compound as defined above, and the superscript "3" does NOT mean a reference. Please delete the introduced reference here.

**Structural basis of specific recognition of PTPδ by SALM5.** In the 1:1 complex of SALM5/PTPδ, the Ig2–Ig3 domains of PTPδ adopt a U-shaped configuration to half wrap the SALM5 Ig domain, with one end of its Ig2 domain docking to the junction of SALM5 LRR and Ig domain. A total of 2540 Å² SASA is buried in the SALM5/PTPδ interface that can be divided into three principal regions, sequentially named site I, II, and III along the primary structure of SALM5 (Fig. 3a). The contacting residues on the SALM5/PTPδ interface are listed in Supplementary Table 3.

In site I, the LRRCT subdomain and loops in the LRR6 and LRR7 of the SALM5 LRR domain form intimate interactions with

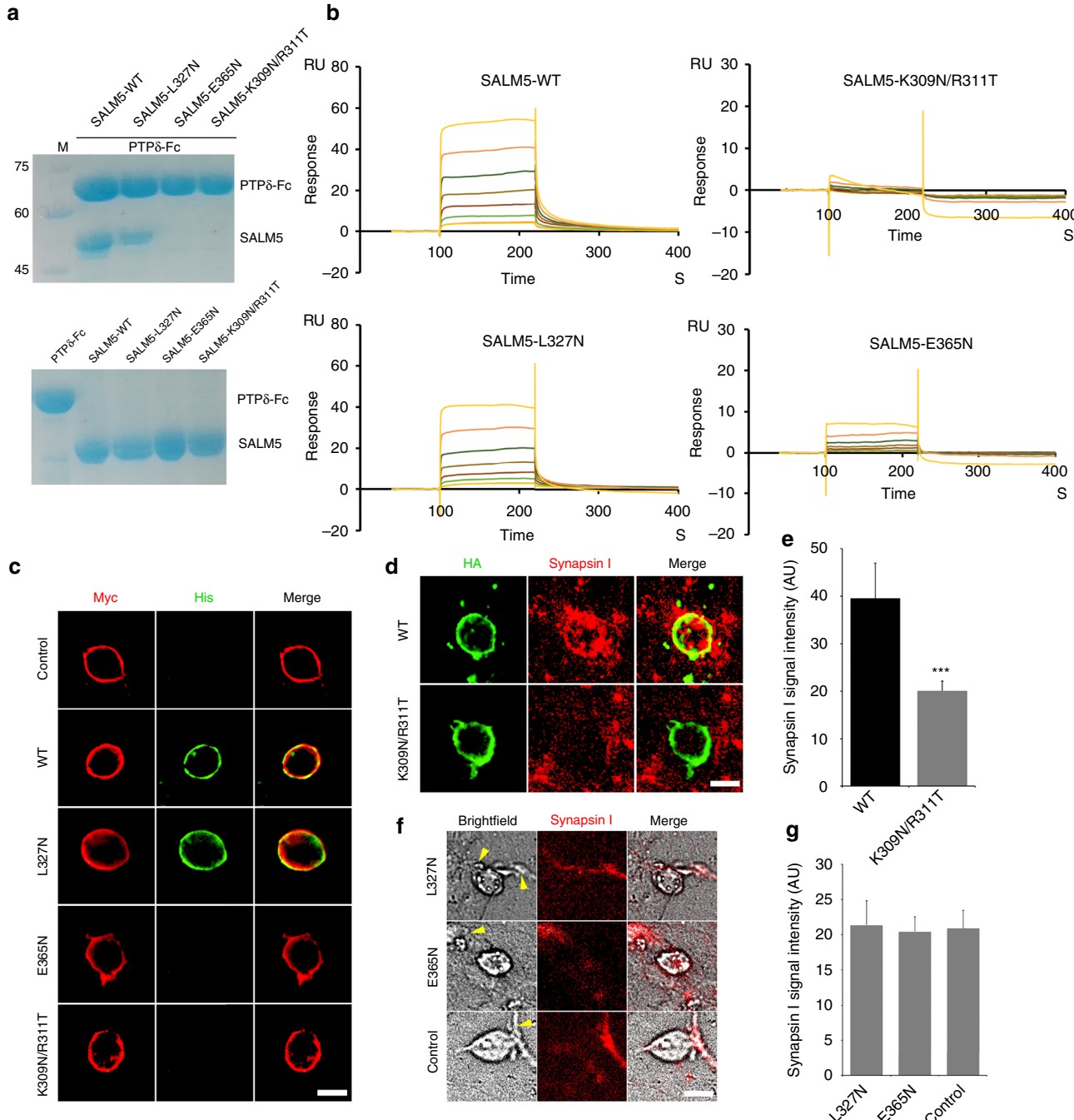

**Fig. 4** Structure-guided biophysical and functional characterizations of the SALM5/PTPδ interaction. **a** Upper panel: SDS-PAGE analysis of SALM5 LRR-Ig (labeled as SALM5 for short) wild-type (WT) and its variants L327N, E365N, and K309N/R311T pulled down by Fc-tagged PTPδ Ig1–3 (labeled as PTPδ-Fc for short); lower panel: SDS-PAGE analysis of proteins used in the pull-down assay. M: molecular weight marker (kDa). **b** SPR sensorgrams of immobilized Fc-tagged PTPδ Ig1–3 (MeA⁻) binding to SALM5 LRR-Ig wild-type (WT) and its variants L327N, E365N, and K309N/R311T. $K_d$ values for WT and L327N are, respectively, 281.8 and 145.3 nM, whereas $K_d$ values for mutants of E365N and K309N/R311T were not determined due to impaired SALM5/PTPδ interactions. **c** Representative images of HEK293T cells displaying Myc-tagged PTPδ Ig1–3 (MeA⁻) incubated with vehicle (control) and recombinant His-tagged proteins of SALM5 LRR-Ig WT and its variants L327N, E365N, and K309N/R311T, and then immunostained with antibodies against Myc-tag (red) and His-tag (green). Scale bar, 20 μm. **d** Representative images of neurons co-cultured with HEK293T cells expressing the WT full-length SALM5 ectodomain (WT) and its mutant K309N/R311T. Scale bar, 20 μm. Immunostaining and synapsin-I intensity measurement were following the protocol used for Fig. 2e. **e** Quantification of the result in **d**. $n = 14$ fields of view, ***$p < 0.005$ for WT vs mutant, Student's $t$ test. **f** Representative images of synapsin-I (red) clustering in neuron co-cultured with HEK293T cells that were transfected with plasmids for empty pDisplay vector, L327N and E365N mutants of the full-length SALM5 ectodomain. Scale bar, 25 μm. Images for HEK293T cells were taken in bright field since mutants L327N and E365N were not expressed and no HA-tagged protein was displayed on the surface of HEK293T cells (Supplementary Fig. 7). In the bright field, the locations of primary neurons are indicated using yellow arrows. **g** Quantification of the result in **d**, $n = 14$ fields of view. Error bars in **e** and **g** present standard error of the mean (S.E.M.)

the β-sheet AFG of PTPδ Ig2 domain, covering ~930 Å$^2$ of SASA in total. The structural chemistry of this site mainly features hydrophobic interactions. SALM5 P212, C246, L249, and W250 form hydrophobic contacts with PTPδ L134, V136, and Y215, which constitutes the hydrophobic core of site I (Fig. 3b). Sheltering this hydrophobic core are hydrogen bonds involved by three basic residues. K188 from SALM5 LRR6 forms a hydrogen bond with the side chain amide of PTPδ Q133. The guanidine group of SALM5 R253 forms bifurcated hydrogen bridges with the side chain amide of Q193 and the main chain carbonyl of V216 in PTPδ. R217 of PTPδ inserts its guanidine group into a hydrophobic cavity formed by SALM5 C246, L249, and L283, and forms a hydrogen bond with the main chain carbonyl of SALM5 E279. Notably, on the edge of site I, R220 from the splicing insert MeB is in proximity, but not forming salt bridge, with SALM5 E280.

In site II, the β-strands βA and βB of SALM5 Ig domain interact with the βC strand and the loop connecting strands βC and βD of PTPδ Ig3 domain, burying a total of ~890 Å$^2$ SASA. The SALM5/PTPδ recognition chemistry in this site features electrostatic complementarity with SALM5 being positively charged and PTPδ negatively charged. The bulged CD loop of PTPδ Ig3 domain, bearing D266 and E270, docks into the positively charged region formed by R291, K309, and R311 of SALM5 (Supplementary Fig. 6a, b). As expected, PTPδ D266 forms a salt bridge with SALM5 R291. Intriguingly, the side chain of PTPδ E270 is sandwiched by side chains of SALM5 K309 and R311, rather than forming salt bridges with them. Instead, PTPδ E270 uses its side chain carboxyl group to form two hydrogen bonds with main chain amide groups of SALM5 A310 and R311. SALM5 R311, in a bifurcated way, hydrogen bonds with the main chain carbonyl group of PTPδ E270 and the aromatic hydroxyl group of Y257 (Fig. 3c). In addition, Site II involves sparse hydrophobic interactions among SALM5 L288 and T290 (C$_\gamma$ atom) and PTPδ M273 and M296.

In Site III, the β-strands βF and βG, as well as CD loop, of SALM5 Ig domain form a long and narrow interface (burying ~825 Å$^2$ SASA) with the β-strands A and B of PTPδ Ig2 domain (Fig. 3d). The interfacial interactions in this site highlight two spots. One is about SALM5 E365. Situated in a position opposite to PTPδ L146, E365 bends its side chain and thus forms a hydrogen bond with PTPδ T144 and a salt bridge with PTPδ K135, which also hydrogen bonds to the main chain carbonyl group of SALM5 N361. The other one involves SALM5 L327, which is on the bulge of CD loop in the SALM5 Ig domain and forms hydrophobic contacts with M130, A148, and D129 (C$_\beta$ atom) of PTPδ. Adjacent to this hydrophobic interaction spot, K326 extends its side chain out to form a salt bridge with PTPδ D154. Noteworthy, a double-alanine mutation of S329 and S360 was reported to weaken SALM5 binding to LAR, PTPδ, and PTPσ[15]. Although both S329 and S360 are hardly involved in direct interaction with PTPδ, their mutation to alanine would impair PTPδ binding in an allosteric manner since they are in proximity to Site III (Supplementary Fig. 6c, d).

**SALM5/PTPδ interaction critical for synaptic differentiation.** In order to decide whether the complex interface presented in crystal structure is energetically important for SALM5/PTPδ interaction, we introduced N-linked glycan wedges to the interface through mutating PTPδ-interacting residues of SALM5 (K309N/R311T, L327N, and E365N) and tested PTPδ-binding capacity of these mutants by pull-down, surface plasmon resonance (SPR) and protein–cell interaction assays. The presence of the glycan wedge, GlcNAc2Man5, in each mutant was confirmed using mass spectrometry (Supplementary Fig. 3). In the pull-

down assay, the wild-type SALM5 LRR-Ig and its variant L327N could be pulled down by Fc-tagged PTPδ Ig1−3 (MeA$^-$), whereas mutants E365N and K309N/R311T could not (Fig. 4a). In the SPR experiments, the wild-type SALM5 LRR-Ig and its variant L327N bound chip-coupled Fc-tagged PTPδ Ig1−3 (MeA$^-$) with approximately close affinities (281.8 vs 145.3 nM). However, mutation of E365N or K309N/R311T abolished SALM5 binding to immobilized PTPδ (Fig. 4b). In line with the pull-down and SPR results, His-tagged SALM5 LRR-Ig mutants E365N and K309N/R311T did not bind to Myc-tagged PTPδ Ig1−3 (MeA$^-$) displayed on the surface of HEK293T cells, whereas the mutant L327N bound to PTPδ-expressing cells as the wild-type SALM5 did (Fig. 4c). In summary, disrupting interactions involved by SALM5 K309, R311, and E365 by introducing a glycan to these sites is lethal to SALM5/PTPδ interaction. Since K309/R311 and E365, respectively, inhabit Site II and Site III of SALM5/PTPδ interface, these data suggested that the two sites are both vital to SALM5/PTPδ interaction. In contrast, introduction of a glycan to SALM5 L327 almost did not alter PTPδ-binding capacity of SALM5. This meant that hydrophobic interactions involved by SALM5 L327 are not critical to SALM5/PTPδ recognition. Meanwhile, the intrinsic flexibility of the Ig CD loop, which SALM5 L327 resides in, likely makes the introduced glycan at this site lack an expected hindrance role as a wedge.

Besides the above protein-binding experiments, we further investigated how mutations of K309N/R311T, L327N, and E365N affected the functionality of SALM5 in inducing presynaptic differentiation. Likely due to accelerated degradation, the mutants of SALM5 ectodomain, L327N and E365N, were not expressed in HEK293T cells as indicated by western blotting and immunostaining experiments (Supplementary Fig. 7). Compared with the wild-type SALM5, the mutant K309N/R311T significantly decreased intensity of synapsin-I clustering in the hippocampal neuron that HEK293T cells formed artificial synapses with (Fig. 4d, e). Incidentally, two mutants L327N and E365N did not lead to synapsin-I clustering (Fig. 4f, g). Combined together, disrupting the SALM5/PTPδ interaction may impair SALM5-dependent synaptic differentiation.

## Discussion

Given that the SALM family members have not been structurally characterized, the SALM5 LRR-Ig structure herein may provide a template for deciphering the structure of other members in this family. T53 and other characteristic residues defining the LRR repeats 1–7 in the SALM5 are conserved or replaced by a hydrophobic residue in other members (Supplementary Fig. 8). Meanwhile, the disulfide bridges distributed in the LRRCT and LRRNT subdomains as well as the Ig domain of SALM5 are absolutely conserved in the whole family (Supplementary Fig. 8). High conservation of these residues suggests that other SALM family members would have a similar topology with SALM5. Furthermore, the residues R110, Q134, and N158 that engage in hydrogen bonds key to SALM5 dimerization in the center of the dimer, as well as the hydrophobic residues on the dimer interface, such as L43, F44, A264, L260, and Y273, are highly conserved in the whole SALM family (Supplementary Fig. 8). This suggests that other SALM family members would also form a dimer with LRR domain packing into an antiparallel side-by-side architecture and that different members may form heteromeric complexes[16]. SALM4 *cis*-inhibiting SALM3-LAR adhesion to suppress excitatory synapse development would likely result from the formation of a side-by-side heterodimer between SALM3 and SALM4[14].

Furthermore, the structure of SALM5/PTPδ may also provide insights into SALM5 binding to LAR and PTPσ, which both have the same folding topology in the first three Ig domains[41]. The

residues Q133, L134, V136, Y215, and R217 of PTPδ important for SALM5 interaction in the Site I, and Y257, D266, and E270 in the Site II, and K135 and T144 in the Site III of the SALM5/PTPδ interface are absolutely or highly conserved in LAR and PTPσ (Supplementary Fig. 9). This implicates that SALM5 would bind to LAR and PTPσ in the same pattern as to PTPδ. Accordingly, in the heterologous synapse assay, disrupting SALM5 dimerization or SALM5/PTPδ interaction may significantly impair presynaptic differentiation, which was indeed mediated by LAR, PTPσ, and PTPδ[16]. Noteworthy to mention, the discrepancy in the primary structures of LAR-RPTPs would determine the difference of SALM5 binding of LAR-RPTPs, such as splice insert-dependent extent[15].

Seemingly contradicting the present complex structure, where the splice insert MeB of PTPδ engages in SALM5 interaction, the MeB insert was reported to inhibit SALM5/PTPδ interaction[15]. In order to quantitatively evaluate how the splice inserts in PTPδ affect SALM5 binding, we carried out SPR experiments (Fig. 5), which clearly showed that the addition of MeB to PTPδ significantly enhanced PTPδ/SALM5 interaction (by 27 or 20 folds, respectively, for the presence or absence of MeA). This implicates that the MeB splice insert in PTPδ is key to SALM5 binding. In comparison, the addition of MeA hardly altered SALM5 binding, indicating that MeA has no much influence on SALM5/PTPδ interaction. Similar to the case of SALM5 binding, the MeB insert in PTPδ is also critical for SALM3 binding[13].

Interestingly, similar to our SALM5/PTPδ complex structure, the structures of PTPδ in complexes with Slitrk1[42], IL1RAPL1, or IL-1RAcP[43] and Slitrk2[44] had the MeB insert in PTPδ (Fig. 6a–e). Superimposition of different ligand-bound PTPδ structures reveals that the MeB insert may act as a universal joint to allow the Ig3 domain to adopt variable orientations (Fig. 6f). Conceivably, the presence of MeB insert would enable PTPδ to switch to an energetically favorable state for ligand interaction. Indeed, among LAR-RPTP variants, PTPδ containing the splice MeB

insert has the highest relative abundance, which suggests that MeB-dependent adhesions may be important for PTPδ functionality in rats and mice[13, 26].

Strikingly, distinct from other PTPδ-binding proteins[42, 44, 43], SALM5 mainly exists as a dimer in solution. As indicated by artificial synapse assay, dimerization is the prerequisite condition for SALM5 functionality in inducing presynaptic differentiation. Combined with mutagenesis disrupting receptor–ligand interaction and cellular assay, the structure of SALM5/PTPδ complex further shows SALM5-induced dimerization of PTPδ is a key step to activate PTPδ for synaptogenesis. Therefore, we here proposed a model in which a dimeric SALM5 induces dimerization and activation of PTPδ for presynaptic differentiation (Supplementary Fig. 10). As previous structural analysis indicated, the antiparallel side-by-side packing architecture of LRR domains in the SALM5 dimer makes two post-synaptic SALM5 molecules in an ideal orientation for cis-interaction. Upon binding to a dimeric SALM5, two PTPδ molecules cluster. Given that the interdomain linkers are likely flexible and the Fn domains of PTPδ may assume a flexion-like configuration similar to its homolog PTPσ[41], it is conceivable that a post-synaptic SALM5 dimer trans-interacts with two presynaptic PTPδ molecules, which would emanate away from the central dyad axis of the SALM5 dimer and then fold back to the presynaptic membrane. Such a bent-back configuration would facilitate fitting the extracellular 11 domains of PTPδ into the narrow synaptic cleft (~240 Å for mammalian excitatory synapses)[22, 49]. Consequently, the intracellular tandem phosphatase domains (D1D2) of PTPδ could be precisely located at a proper position, although the precise distance between two phosphatase domains is not known (Supplementary Fig. 10a). Given that the dominant isoform of PTPδ in the developing brain contains only four Fn domains[43], such a model needs a modification based on flexibility of interdomain linkers in PTPδ and SALM5 (Supplementary Fig. 10b).

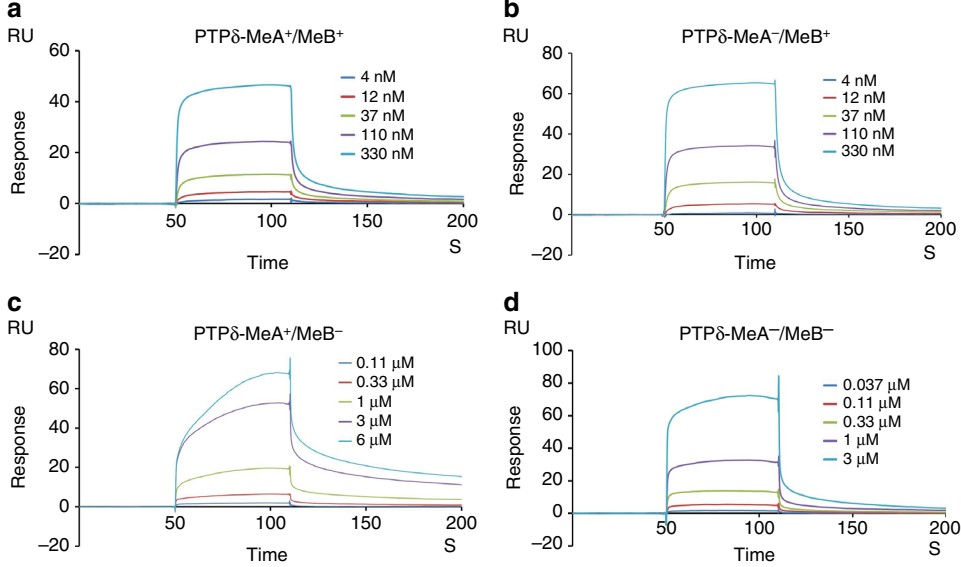

**Fig. 5** SPR analysis of SALM5 interacting with PTPδ splice code variants. **a–d** are SPR sensorgrams of immobilized human SALM5 LRR-Ig binding to human PTPδ Ig1–3 WT (**a**) as well as vairants MeA⁻ (**b**), MeB⁻ (**c**) and MeA⁻/MeB⁻ (**d**). For clarity, human PTPδ Ig1–3 is labeled as "PTPδ." For each analyte, five concentrations were used as indicated. The WT human PTPδ Ig1–3 and its variants MeA⁻, MeB⁻, and MeA⁻/MeB⁻ bind to SALM5 with affinities of 66.0, 99.9, 1780.0, and 1866.0 nM, respectively. The binding affinity for SALM5 and PTPδ Ig1–3 (MeA⁻) determined here (with SALM5 immobilized) is slightly higher than the reversed interaction (with PTPδ immobilized) shown in Fig. 4b (99.9 vs 281.8 nM), likely due to avidity effect of the SALM5 dimer immobilized on the chip. Different from the previous L-cell aggregation assay-based report[15] that MeB in PTPδ additively suppressed PTPδ binding to SALM5, our SPR experiments revealed that the addition of MeB to PTPδ significantly enhanced its binding to SALM5. This difference might derive from the different methods employed to assess SALM5/PTPδ interaction

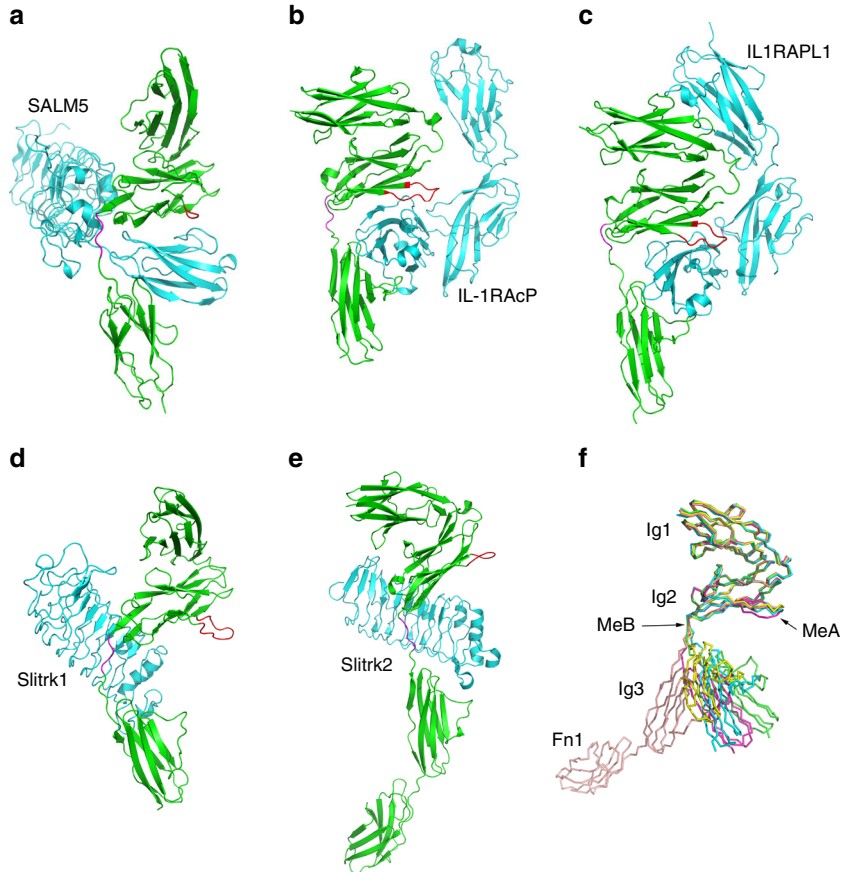

**Fig. 6** Comparison of PTPδ in complexes with different ligands. **a–e** Ribbon representations of PTPδ in complexes with SALM5 (this paper; PDB code, 5XNP), IL-1RAcP (4YFD), IL1RAPL1 (4YH7, Fn domains not shown), Slitrk1 (4RCA), and Slitrk2 (4Y61). PTPδ and PTPσ are colored in green with MeA splice insert in red and MeB in pink. **f** Superimposition of PTPδ bound with different ligands: SALM5 (green), IL1RAPL1 (cyan), IL-1RAcP (yellow), Slitrk1 (pink), and Slitrk2 (salmon). PTPδ bound with Slitrk2 has an extra domain, Fn1, represented in the structure

Since the key residues in PTPδ for SALM5 binding are highly conserved among the LAR-RPTP family that may share a similar topology, the above mechanism model would be suitable for SALM5 activating other LAR-RPTP family members. Universality of such a model may explain the recent report that a double mutation of S329 and S360 not only impaired SALM5 binding to the whole LAR-RPTP family, but also weakened presynaptic differentiation governed by them[15]. However, this model could not cover any cases, given that some isoforms of PTPδ contain only one Ig domain[50], that SALM5 may engage in intercellular adhesion[16], and that PTPδ could be located at post-synaptic sites[51]. Since PTPδ/IL1RAPL1 in *cis*-interaction may antagonize IL1RAPL1 function[51], we propose that *cis*-interaction between SALM5 and PTPδ (or truncated isoforms) on pre- or post-synaptic membrane might competitively regulate SALM5-mediated synaptic differentiation.

Different from other RPTPs, *cis*-dimerization or oligomerization of which negatively regulates their catalytic activity[52–55], dimerization of LAR-RPTPs may couple with phosphatase activity for an activating regulation[56]. Crystallographic studies have suggested that LAR-RPTP tandem phosphatase domains are monomeric in solution[40, 57, 58]. This means that extracellular ligand binding would be critical for LAR-RPTP oligomerization and activation. Therefore, our SALM5-mediated PTPδ dimerization model sets a point for unraveling LAR-RPTP transmembrane signaling mechanism. A caveat about this model is that PTPσ was suggested to exist as a dimer on the cell[59]. Due to lack of direct biochemical evidence, we could not rule out the

possibility that PTPδ is a dimer on the cell membrane as PTPσ. In case of this possibility, binding to a dimeric SALM5 would facilitate remodeling of LAR-RPTPs or seduce a conformational modulation so that they could be activated for downstream signaling in synaptogenesis.

Another extreme possibility is that SALM5-mediated PTPδ dimerization would not couple with intracellular phosphatase activation. For this possibility, it is still conceivable that SALM5-induced dimeric clustering of PTPδ would result in enrichment of PTPδ in local milieu for enhancing PTPδ signaling efficiency. Indeed, RPTP clustering would lead to a redistribution of tyrosine phosphatase activity at the plasma membrane[22]. A good example for this point is HSPGs (heparan sulfate proteoglycans), which induce tetrameric clustering of PTPσ that leads to the promotion of neurite outgrowth[60]. The NT3 neurotrophin may also induce formation of dimeric or higher-order RPTPσ arrays through binding to the TrkC/PTPσ complex, thus enhancing the synaptic organizing function of TrkC/PTPσ in rat hippocampal neurons[61]. From this point, the SALM5-induced PTPδ dimerization still has biological significance.HSPGs (heparan sulfate proteoglycans).

In conclusion, we have determined the crystal structure of a 2:2 SALM5/PTPδ heterotetramer that yields insights into how a SALM5 dimer induces *cis*-dimerization of PTPδ for their function in presynaptic differentiation. No matter which mechanism will underlie SALM5-mediated LAR-RPTP transmembrane signaling, it is conclusive that SALM5-induced dimerization or precise repositioning of LAR-RPTPs is vital for the functionality of SALM5 in synaptogenesis. Future structural, biochemical, and functional

studies on the full-length LAR-RPTPs in the absence or presence of SALM5 will be crucial to further explain how dimerization of SALM5 is coupled with their intracellular tyrosine phosphatase activity for synaptogenesis.

## Methods

**Baculovirus recombination and protein preparation**. Proteins used for crystallization, gel filtration assay, mass spectrometry, AUC study, pull-down assay, surface plasmon resonance analysis, cell surface-binding assay, and polyantibody generation were expressed in the BacMam system[62]. cDNAs encoding human SALM5 LRR-Ig (residues 18–374) or its mutants (R110N/E160S, K309N/R311, L327N, and E365N), human SALM5 ectodomain (residues 18–526), human PTPδ Ig1–3 (MeA⁻) (residues 21–320 with ESIGGTPIR 181–189 deleted; also labeled as MeA⁻/MeB⁺), human PTPδ Ig1–3 (MeB⁻) (residues 21–320 with ELRE 227–230 deleted; also labeled as MeA⁺/MeB⁻), human PTPδ Ig1–3 (MeA⁻/MeB⁻) (residues 21–320 with both residues 181–189 and 227–230 deleted) and human PTPδ Ig1–3 WT (with MeA and MeB remained; also labeled as MeA⁺/MeB⁺) were amplified by a single-step PCR or a two-step overlapping PCR (see Supplementary Table 1 for primer sequences) and cloned into a modified BacMam vector with N-terminal *Gaussia princeps* luciferase signal sequence and C-terminal 7× His-tag[62]. For producing Fc-tagged proteins, human PTPδ Ig1–3 (MeA⁻) gene was also cloned into a BacMam vector with C-terminal human IgG1 Fc plus 7× His-tag. For each construct, the resulted transfer plasmid was used to transfect *Sf9* insect cells in the presence of BacVector-3000 baculovirus DNA (EMD Biosciences). Seven days after transfection, low-titer baculovirus stock was harvested and sequentially used to infect insect cells for amplification. For protein expression, the amplified high-titer baculoviruses were used to transduce HEK293 GnTI⁻ cells[63] at a density of $2 \times 10^6$ cells per ml. The cell culture was further maintained in suspension for 72 h at 37 °C in a humidified incubator supplemented with 5% $CO_2$. Cells were removed by centrifugation at $3000 \times g$ for 15 min, and the conditioned medium was concentrated and completely buffer exchanged into high-salt HEPES-buffered saline (HBS, 10 mM HEPES pH 7.5, 500 mM NaCl). Interest proteins were captured using Ni²⁺-NTA agarose (Invitrogen) and in gradient eluted with HBS supplemented with different concentrations of imidazole. The eluate was confirmed using SDS-PAGE and then subjected to a gel-filtration column (Superdex 200 Increase, GE) pre-equilibrated and washed with HBS. The fractions were pooled and concentrated to $8-10$ mg ml⁻¹ for crystallization or to a proper concentration for biochemical and biophysical assays. For detecting complex assembly, the purified SALM5 and PTPδ proteins were mixed for 1-h incubation at 4 °C and then subjected to gel-filtration analysis performed on the same column. Samples from peak fractions were analyzed using SDS-PAGE and visualized by Coomassie blue staining. All the recombinant proteins were purified sequentially using Ni²⁺-NTA affinity and gel-filtration chromatography. Except that used for crystallization, all the purified recombinant proteins were not applied to glycan trimming.

**Crystallization and cryoprotection**. The purified human SALM5 LRR-Ig alone at 8 mg ml⁻¹ or its complex with human PTPδ Ig1–3 (MeA⁻) at 10 mg ml⁻¹ was trimmed using Endo H (New England Labs) for glycan removal. Protein crystallization was performed at 20 °C using the vapor diffusion sitting-drop method. The drops were set up with 0.5 μl protein in mixture with 0.5 μl reservoir solution. Diamond-like SALM5 LRR-Ig crystals grew from drops equilibrated against reservoir solution containing 2% (w/v) polyethylene glycol (PEG) 3000, 0.1 M sodium acetate (pH 4.5), and 0.2 M Li₂SO₄. The complex crystals with a cubic shape were observed after 3 weeks from reservoir solution containing 0.76 M potassium sodium tartrate and 0.1 M MES (pH 6.0). Crystals were flash-frozen in liquid N₂, followed by soaking in the reservoir solutions supplemented with 26% (v/v) ethylene glycol for SALM5 LRR-Ig crystals and 20% (v/v) ethylene glycol for complex crystals. To prepare a heavy atom derivative, SALM5 LRR-Ig crystals were soaked in the cryoprotectant buffer containing 0.2 M NaI for 3 min before being flash cooled.

**Structure determination and refinement**. All the diffraction data sets were collected at Shanghai Synchrotron Radiation Facility (SSRF) and processed with HKL2000[64]. On the BL17U1 beamline, two sets of high-redundancy data were collected at the wavelength of 1.54719 Å for native and NaI-derivative SALM5 LRR-Ig crystals. The two processed data sets were subjected to the AutoSol program in PHENIX[65] for locating iodine atoms and calculating initial phases using the single isomorphous replacement with anomalous scattering (SIRAS) method. Some LRRs of SALM5 could be discerned in the electron density map calculated from initial SIRAS phases. The initial model was improved by the AutoBuild program in PHENIX that generated a model with 60% of SALM5 LRR domain residues assigned. The model was then completed by iterative manual building in COOT[66] and refinement in PHENIX against a 2.8 Å resolution data collected on the BL19U1 beamline with a wavelength of 0.97853 Å. Diffraction data for crystal of human SALM5 LRR-Ig in complex with PTPδ Ig1–3 (MeA⁻) were also collected on the BL19U1 beamline with the same wavelength. The complex structure was solved by the molecular replacement (MR) method using the PHASER program in the CCP4 suite[67]. The N-terminal two Ig domains of PTPδ from IL1RAPL1/PTPδ complex (PDB code 4YH7)[43], together with the structure of SALM5 LRR domain,

was used as the search model. A sigma-A weighted $2F_o–F_c$ electron density map calculated from the MR solution was clear enough to manually locate the third Ig domain of PTPδ and the Ig domain of SALM5. The resulting complex model was iteratively refined in PHENIX and manually built in COOT. Guided by a sigma-A weighted $F_o–F_c$ electron density map, waters, glycans, ions, and buffer molecules were cautiously integrated into the final models for SALM5 LRR alone and its complex with PTPδ Ig1–3 (MeA⁻). The stereochemistry of the final model was assessed by the program Molprobity in PHENIX. Percentages of residues in the favored regions of the Ramachandran plot are 98.56 and 97.76% for free SALM5 and SALM5/PTPδ structures, and no residues are distributed in the outlier region. Data collection, phasing, and refinement statistics are summarized in Table 1. All structural figures are prepared using the program PyMol (Schrödinger, LLC).

**LC-MS/MS analysis**. The nano-LC-MS/MS experiments were performed using a LTQ orbitrap velos pro mass spectrometer (Thermo). Tryptic peptides were applied onto an EASY nano-LC system following the manufacturer's instructions. Each elute was then subjected into a C18 reverse phase column (75 μm i.d., 10 cm long, 3 μm resin from Thermo). The peptide mixtures were eluted with a 0–40% gradient from buffer A (0.1% formic acid) to buffer B (0.1% formic acid in acetonitrile) over 30 min and were then online detected in mass spectrometer using a data-dependent TOP15 method. The general mass spectrometric conditions were spray voltage, 2.2 kV; no sheath and auxiliary gas flow; ion transfer tube temperature, 250 °C. For MS/MS (MS2), normalized collision energy using wide-band activation mode was 35%, and ion selection threshold was 1000 counts. An activation $q$ of 0.25 and activation time of 30 ms were applied in MS2 acquisitions. The mass spectrometer was operated in positive ion mode, and a data-dependent automatic switch was employed between MS and MS2 acquisitions. For each cycle, one full MS scan in the Orbitrap was followed by 15 MS2 in the LTQ on the ten most intense ions. Each precursor ion selected for MS2 was dynamically excluded for subsequent LC-MS runs with 30 s exclusion duration. The LC-MS/MS data were submitted to database searching against the human SALM5 sequences with mutations R110N/E160S, K309N/R311T, L327N, and E365N using the software Byonic™. Mass tolerances were 10 ppm and 0.6 Da for the precursor and fragment ions, respectively. Enzyme specificity was set as KR/P, and a maximum of two missed cleavages was allowed. Asparagine glycosylation, cysteine carbamidomethylation, and methionine oxidation were set as variable modifications.

**Analytical ultracentrifugation**. Sedimentation velocity experiments were performed on a Beckman XL-I analytical ultracentrifuge at 20 °C. Protein samples were diluted using HBS buffer to 400 μl at an A280 nm absorption of about 0.8. Samples were loaded into a conventional double-sector quartz cell and mounted in a Beckman four-hole An-60 Ti rotor. Data were collected at 50,000 rpm at a wavelength of 280 nm for 7 h. Interference sedimentation coefficient distributions were calculated from the sedimentation velocity data using the SEDFIT software.

**Pull-down assay**. Qualitative analysis of SALM5/PTPδ interaction was performed by pull-down assay with the C-terminally Fc-tagged PTPδ Ig1–3 (MeA⁻). The Ni²⁺-NTA affinity-purified SALM5 LRR-Ig and its mutants, K309N/R311T, L327N, and E365N, were respectively mixed with PTPδ Ig1–3 (MeA⁻)-Fc at a molar ratio of 1.5:1, and then immobilized with Protein G Sepharose resin (Solabio, China) by rotation at room temperature for 2 h. After intensively washing with phosphate-buffered saline (PBS), the resin-bound protein complexes were eluted by 0.1 M glycine (pH 2.7) and immediately resolved by SDS-PAGE and Coomassie blue staining.

**SPR analysis**. All the SPR experiments were carried out on BiacoreT200 (GE Healthcare) instrument at 25 °C in a running buffer containing 10 mM HEPES (pH 7.4), 150 mM NaCl, 3 mM EDTA, and 0.05% surfactant P20. For measuring binding affinities for PTPδ interacting with SALM5 and its mutants, the purified Fc-tagged PTPδ Ig1–3 (MeA⁻) was immobilized to the surface of a CM5 sensor chip using the amine-coupling method at a density of about 500 resonance units (RU), while the wild-type SALM5 LRR-Ig and its variants K309N/R311T, L327N, and E365N were, respectively, injected as the analytes in serial twofold dilutions with the concentrations ranging from 23.43 to 1500 nM. For determining binding affinities of SALM5 with different splice code variants of PTPδ, the His-tagged SALM5 LRR-Ig was immobilized on a CM5 chip, while the wild-type His-tagged PTPδ Ig1–3 and its variants MeA⁻, MeB⁻, and MeA⁻/MeB⁻ were run as the analytes in serial threefold dilutions with concentrations ranging from 2.74 to 6000 nM. Generally, each analyte sample was injected over the chip at a flowrate of 30 μl min⁻¹ for a 120 s or 60 s association and then the running buffer was switched over the chip to allow the analytes to undergo a 180 s or 90 s dissociation and the signal to return back to baseline. No regeneration of the chip surface was required between different analyte injections, since injection of the running buffer could bring the signal back to baseline. All SPR data were analyzed assuming the Langmuir model with a ligand to analyte molar ratio of 1:1.

**Cell surface-binding assay**. Human PTPδ Ig1–3 (MeA⁻) coding sequence was amplified PCR (See Supplementary Table 1 for primer sequences) and cloned into pDisplay (Invitrogen) in frame with the Igκ chain leader peptide and Myc epitope

coding sequences on the vector. In the presence of Lipofectamine 3000 (Invitrogen), the constructed plasmid was used to transfect HEK293T cells seeded on a 96-well plate. Forty-eight hours after transfection, the purified SALM5 LRR-Ig WT and mutants, K309N/R311T, L327N, and E365N, at a concentration of 10 μg ml$^{-1}$, were respectively added to PTPδ-expressing live HEK293T cells and incubated for 2 h at 37 °C. HBS was added to cells as a vehicle control. After incubation, cells were washed using PBS twice, fixed with 4% (w/v) paraformaldehyde and incubated with rabbit anti-Myc tag (Proteintech) and mouse anti-His-tag (Biodragon, China) antibodies in the presence of 3% (w/v) BSA. After removing the primary antibodies, cells were washed with PBS twice, and the secondary antibodies, PE-conjugated goat anti-rabbit IgG (Transgen, China) and Alexa Fluro 488-conjugated goat anti-mouse IgG (Jackson) were added for immunostaining PTPδ and SALM5.

**Chemical cross-linking assay.** Human SALM5 LRR-Ig and its mutant, R110N/E160S, were respectively incubated with bis (sulfosuccinimidyl) suberate (BS$^3$) at a molar ratio of 1:100. After incubation at room temperature for 30 min, 50 mM Tris-HCl (pH 7.5) was used to quench the reaction. The cross-linking mixture was applied to SDS-PAGE analysis, followed by Coomassie blue staining.

**Generation of polyclonal antibody against human SALM5.** Approval for use of animals was provided by the Animal Care and Use Committee of Peking University Health Science Center (Beijing, China). Mouse experiments were conducted in conformity with institutional guide for the care and use of laboratory animals. Three 8-week-old female BALB/c mice were immunized subcutaneously with 200 μg of purified human SALM5 LRR-Ig protein per injection with complete Freund's adjuvant for the first dose and incomplete Freund's adjuvant for subsequent doses. The mice received two booster injections at 10-day intervals. Blood was taken for sera analysis from mice harvested 4 days after the final boost. Sera were obtained from blood samples by centrifugation and applied to ELISA evaluation of anti-SALM5 polyclonal antibodies.

**Heterologous synapse formation assay.** cDNAs encoding human SALM5 ectodomain (residues 18–526) and its variants, K309N/R311T, L327N, and E365N, were amplified using one-step or two-step overlapping PCR (See Supplementary Table 1 for primer sequences) and cloned to pDisplay (Invitrogen) in frame with N-terminal hemagglutinin A (HA) epitope and C-terminal Myc epitope. The constructed expression plasmids were transfected to HEK293T cells using Lipofectamine 3000 transfection reagent (Invitrogen). After 2 days of culture, the transfected cells were harvested for western blot analysis using anti-SALM5 polyclonal antibodies from immunized mice. Primary hippocampal cultures were prepared as described in the reference[68]. Briefly, embryonic rat hippocampi were dissected out of E18 embryos, trypsinized and triturated to single-cell suspension. The resulted cells were seeded on a 96-well plate coated with poly-L-lysine and cultured in Neurobasal A (Invitrogen) supplemented with 2% B-27 supplements, 100 U ml$^{-1}$ penicillin and 100 μg ml$^{-1}$ streptomycin. At DIV9/10, wild or mutant SALM5 ectodomain-expressing HEK293T cells were harvested by centrifugation, dispersed in Neurobasal A medium, and then plated into cultured hippocampal neurons at a density of ~5000 HEK293T cells per well. After 48 h of co-culture, cells were fixed with 4% paraformaldehyde for 1 h at 37 °C, permeabilized with 0.5% (v/v) Triton X-100 in PBS for 30 min at 4 °C and blocked with 5% (w/v) BSA for 2 h at 37 °C. For immunostaining SALM5, fixed cells were incubated with mouse anti-HA monoclonal antibody (Biodragon, 1:500) for 2 h at 37 °C, washed thrice with PBS and then incubated with Alexa Fluro 488-conjugated goat anti-mouse IgG antibody for 40 min at 37 °C. For detecting synapsin-I, cells were sequentially incubated with rabbit anti-synapsin-I (Merck-Millipore, 1:500) and PE-conjugated goat anti-rabbit IgG antibodies following the procedure as used for SALM5 immunostaining. Prior to image acquisition, cells were washed thrice with PBS to remove non-specifically bound antibodies.

**Image acquisition and processing.** All images were acquired with Operetta High-Content Imaging System (PerkinElmer), and collected from at least two separate cell cultures. For cell surface binding and heterologous synapse formation assays, images were respectively taken under a set of constant conditions including z value and zoom setting. For both assays, Alexa 488 and Alexa 594 channels were selected with the excitation ($\lambda_{ex}$) and emission ($\lambda_{em}$) filter sets to detect Alexa Fluro 488 ($\lambda_{ex}$: 460–490 nm; $\lambda_{em}$: 500–550 nm) and PE ($\lambda_{ex}$: 560–580 nm; $\lambda_{em}$: 590–640 nm). Fields of view were randomly selected for each well of fixed culture. All quantitative measurements on fluorescence intensity were performed with the web-based software Columbus. Data were represented as mean ± standard error of the mean (S.E.M.). Statistical comparisons were evaluated with Microsoft Excel using two-tailed Student's $t$ test. A value of $p < 0.05$ was considered statistically significant.

**Data availablity.** The authors declare that the main data supporting the findings of this study are available within the article and its Supplementray Information file. Uncropped images of gels and blots that include the marker lanes are available in Supplementray Fig. 11. Other data are available from the corresponding author on reasonable request. Coordinates and structure factors for human SALM5 LRR-Ig alone and in complex with human PTPδ Ig1–3 (MeA$^-$) have been deposited in the Protein Data Bank (PDB) under accession numbers 5XNQ and 5XNP, respectively.

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

## Acknowledgements

This work was supported by grants to H.L. from the China "Thousand Talents Program" and the "Hundred Talents Program" of Peking University. We are grateful to Dr Bo Xu for assistance with operating Operetta High-Content Imaging System, Dr Yaxin Lou for mass spectroscopy, Dr Xiaoxia Yu for AUC and Dr Jing Wang for SPR experiments. We also thank the staffs from BL17U1 and BL19U1 beamlines at Shanghai Synchrotron Radiation Facility for assistance during data collection.

## Author contributions

H.L. conceived and supervised the whole project. Z.L. designed and performed molecular cloning, protein preparation and crystallization, crystal handling, cross-linking reaction, and heterologous synapse formation assay. Z.L. and J.L. performed SPR analysis and protein–cell interaction experiments. Z.L. and H.D. generated polyclonal antibody. Z.L., F.X., and H.L. performed data collection and structure determination. Z.L. and H.L. analyzed the data and wrote the manuscript.

## Additional information

**Competing interests:** The authors declare no competing financial interests.

