## [Peer Review File · Nature Communications]

Reviewers' comments:

Reviewer #1 (Remarks to the Author):

This study by Lin et al. reports the crystal structure of a 2:2 SALM5/PTPd heterotetramer. The authors find that both the interactions between two LRRs as well as SALM5-PTPd are important for the formation of the tetramer as well as the induction of presynaptic differentiation in contacting axons.

Given the reported association of SALM5 with schizophrenia and autism spectrum disorders, and considering the fact that this is the first crystal structure of the SALM/Lrfr family proteins that proposes an important hypothesis that induced PTPd dimerization would be important for presynaptic activation of tyrosine phosphatase activity and presynaptic differentiation, these findings are quite interesting and represent a significant step forward in the field.

Major comments:

1. In previous reports (Choi et al, 2016, Scientific Report), the ecto domain of SALM5 strongly binds MeB - forms of LAR-RPTPs (LAR, PTPd, and PTPs). In addition, insertion of MeB dramatically reduces complex formation. The authors solve the structure of PTPd MeB+ in complex with SALM5. Since the ligands of LAR-RPTPs showed the splice code specificity, the investigation of splice code selectivity of SALM5 is important to understand and generalize the ligand specificity of LAR-RPTPs by modulating splice code. Hence, the author should provide the binding affinity comparison in terms of splice code variants.

2. The authors suggest the trans-synaptic interaction between PTPd and SALM5 in the manuscript. However, the SALM5 could be present at both pre- and postsynaptic sites, considering the reported homophilic and intercellular adhesion of SALM5. In addition, Motani et al. (J Neurosci 2017) have recently suggested the possibility that PTPd could be located at postsynaptic sites. Therefore, the possibility of SALM5-PTPd interaction occurring in a cis manner at both pre- and postsynaptic should be discussed.

3. In Supp Figure S3b, the intensity of SALM5 R110N/E160S seems not sufficient to figure out the exact elution volume of the peak (blue solid line). Could the author increase the concentration of the mutant and carefully compare with wildtype? If the elution volume is similar with that of wild-type protein, the mutant might be a dimer in solution.

4. Figure 2d should show similar amounts of WT and mutant SALM5 proteins.

Minor comments:

1. In Figure 4c,d,e, related text and figure labeling for specific mutations (L317N, E365N, and K309N/R311T) do not match with each other. In addition, the authors are showing in Figure 4d the images that the authors have already shown in Figure 2 again. These apparent mistakes should be corrected.

2. The characteristic of the SALM5-PTPd interaction, according to the authors, forces the PTPd molecule to conform a 'V' shape regarding its Ig domains, from which the FNII-like domains then round back to the presynapse. Naturally, the authors assume that for that reach to occur, PTPd will have to display the entire eight FN-like domain chain, as the reach from the presynapse will have to be ~162Å (Supp Figure 9). This, then, raises questions as to the biological significance of this interaction, because the dominant isoform of PTPd in the brain contains only four FNIII-like domains.

This should be considered in interpreting the results and drawing models for Supp Figure 9.

3. In addition to Kwon et al, the Woo et al paper (Nat Neurosci, 2009) should also be cited.
4. Page3, line 55; the author should provide the full name when the term appears for the first time in the manuscript; CNS, Central Nerve System.
5. Line 177. I wonder the "LRRNT" should be "LRRCT".
6. Line 251. Typo. Monomeric.
7. Line378. SALM5 family should be changed to SALM family.

Reviewer #2 (Remarks to the Author):

This paper, by Lin et al, describes the first crystal structures of the ectodomain region of a SALM-family protein, SALM5, both in isolation and in complex with the LAR-family receptor tyrosine phosphatase PTP δ . SALM5 is thought to impact neuronal differentiation by interacting with LAR-RPTPs, and the current paper begins to reveal the corresponding mechanism in atomic detail. The 2.8Å and 3.7Å structures of SALM5 and its complex with PTP δ appear to be of high quality (although free R factors appear somewhat high). The complex structure shows that SALM5 is a strongly associating rod-shaped dimer, and the rod-shaped SALM5 dimer is bound at either end by a single PTP δ molecule, thus "dimerizing" PTP δ , with an intervening space of about 40Å (the length of the SALM5 rod). The paper validates this interpretation with site-directed mutagenesis and solution biophysical measurements, and conservation suggests other SALM-family members may function similarly. Functional experiments also confirm the model, showing that formation of the crystallographically observed complex is necessary to induce the recruitment of synapsin.

Overall, this is an excellent study on an interesting subject, which I think to be appropriate for publication in Nature Communications.

Reviewer #3 (Remarks to the Author):

Manuscript: # 133492

Authors: Zhaohan Lin, Jianmei Liu, Huandi Ding, Fei Xu, Heli Liu

Title of manuscript: Structural basis of SALM5-induced PTP δ dimerization for synaptic differentiation

This manuscript presents the crystal structures of human SALM5 LRR-Ig alone and in complex with human PTP δ Ig1-3 domains with the mini-exon B insert. SALM5 and PTP δ are two synaptic adhesion molecules that play a role in synapse development, and are associated with neuropsychiatric diseases.

To substantiate the structural findings, the authors carry out biochemical experiments including structure-guided mutagenesis and heterologous synapse formation assays are carried out.

The key findings that this manuscript would like to present are the following:

- the structures of SALM5 (2.8 Å) and its complex with the fragment human PTP δ Ig1-3 (3.7 Å)
- while the Ig domain in SALM5 is likely flexible, in the complex with PTP δ Ig1-3, the Ig2 and Ig3 from PTP δ wrap around the SALM5 Ig domain fixing the latter's conformation.
- SALM5 exists as a dimer in solution, mediated by the LRR domains which pack in an antiparallel

fashion

- dimerization of SALM5 is prerequisite for its functionality in inducing synaptic differentiation.
- SALM5 directly induces cis-dimerization of LAR-RPTPs into higher-order signaling assembly which mediate the clustering of synapsin I (as a readout for presynaptic differentiation).

Significance: The mechanism of SALM5 function and its interaction with PTP δ has important implications for how these synaptic adhesion molecules can promote the formation of synapses.

The structural work is beautiful. Concern, however, exists that important controls in the biochemical and cell-based assays are not included in the manuscript (as discussed below) which makes it difficult to use these experimental data to solidly support the (structural and functional) conclusions from the manuscript that 1) SALM5 is dimeric in solution, 2) the interaction between SALM5-PTP δ Ig1-Ig3 as seen in the crystal structure represents interactions in the physiological complex, and 3) if you disrupt SALM5-PTP δ Ig1-Ig3 interaction, you prevent synapsin I clustering (as a read-out for presynaptic differentiation).

Major comments:

1. Results section p. 8 lines 155-157

"... PTP δ -bound SALM5 structures unanimously revealed that two SALM5 molecules form a diad dimer through their LRR domains packing in an antiparallel side-by-side manner."

The supplemental figure S1c and S1d does not suffice as evidence for the statement above, because the reader cannot easily see how different or similar the SALM5 dimer is alone versus in the complex with PTP δ Ig1-Ig3.

Please provide a figure whereby the SALM5 dimer (from the structure of the SALM5+PTP δ Ig1-Ig3 complex) is superimposed on top of the SALM5 dimer (from the structure of SALM5 alone).

This is crucial because the SALM5 dimer in the SALM5+PTP δ Ig1-Ig3 complex is apparently generated by non-crystallographic symmetry, however, the SALM5 dimer in the structure of SALM5 alone is generated by crystallographic symmetric.

2. Results section p. 8 lines 157 – 161.

"The LRR domains of two SALM5 molecules constitute a concave platform as the core of the dimer, with approximate dimensions of 56 Å × 68Å. In each SALM5 protomer, the Ig domain keeps its long axis perpendicular to the platform; the two Ig domains in the SALM5 dimer are in parallel with each other and no contacts are observed between them (Figure 1d)."

Please revise the text to make clear that you are referring to SALM5 as seen in the complex with PTP δ Ig1-Ig3. This is important because in absence of PTP δ Ig1-Ig3, the Ig domain in SALM5 is not visible.

3. In the legend of Fig. 1d, please indicate in the text that you are showing SALM5 as it is seen in the context of the complex with PTP δ Ig1-Ig3.

For each figure legend throughout the manuscript indicate whether you are showing the structure of SALM5 alone (2.8 Ang. structure) or SALM5 extracted from the complex (3.7 Ang. structure).

4. Results section p. 12 line 249

"Since SALM5 mainly exists in solution as a dimer,...."

Please show experimental evidence for this statement, for example, running the SALM5 sample on a size exclusion column in parallel with gel filtration calibration protein standards or analytical

centrifugation studies.

Comparison of Fig. S3a and S3b does not suffice, because calibration markers for the chromatographic runs in Fig. S3 are lacking, so it is not possible to correlate elution volume with estimated molecular weight. Calibration markers are also needed to be able to interpret if a difference between 13.8 ml and 14.3 ml is significant between wild type SALM5 and SALM5 R110N/E160S (Fig. S3a and S3b), suggesting dimeric versus monomeric states respectively.

5. SALM5 R110N/E160S (Fig. 2d) does not appear more heavily glycosylated, which would be expected, if the glycan wedge was formed in order to disrupt dimer formation of SALM5.

Please demonstrate that R110N/E160S are N-linked glycosylated or explain in the text how the exact protein samples run on the gel filtration columns in Fig. S3b were expressed and purified. Please explain in the text if the proteins used for Fig. S3 were expressed in HEK293 GnTI- cells and EndoH treated (i.e. lacking glycosylation)? Can you rule out that R110N/E160S cause folding defects?

The markers on the SDS-PAGE gel inserts for Fig. S3a and S3b seem different. Is this correct?

6. Because the cross-linked SALM5 is a minor band in Fig. 2d, it is important to load the same amount of protein for the mutant SALM5 R110N/E160S. Approximately 4x more wild type SALM5 is loaded on the gel compared to the mutant SALM5 R110N/E160S, so a cross-linked species might not been seen because insufficient sample is loaded for the mutant SALM5 R110N/E160S.

Please justify or re-run the SDS-PAGE gel.

7. Results section p. 12 line 259

"The monomeric SALM5 LRR-Ig is still capable of binding to PTP δ ,..."

Unless the text can be revised to accommodate points 4, 5 and 6 above, the text should be corrected to read "SALM5 R110N/E160S, which we believe is possibly monomeric, is still capable of binding to PTP δ ,..."

8. Fig. 2e. Please display a brightfield picture showing the image of the HEK293T cell and the neuron that are co-cultured and being imaged. This is important in order to demonstrate that synapsin I is being clustered on a neuron. In the current image, it is not clear where the neuron boundary is.

Please also indicate in the legend of Fig. 2e which proteins are HA-tagged.

9. Results section p. 14 lines 332 – 333, line 345 and lines 349-353.

" we introduced N-linked glycan wedges to the interface through mutating PTP δ -interacting residues of SALM5 (K309N/R311T L327N and E365N)"

" In summary, disrupting interactions involved by SALM5 K309, R311 and E365 by introducing a glycan to these sites, ..."

"In contrast, introduction of a glycan to SALM5 L327 almost didn't alter PTP δ -binding capacity of SALM5. This meant that hydrophobic interactions involved by SALM5 L327 are not critical to SALM5/PTP δ recognition. Meanwhile, the intrinsic flexibility of the Ig CD loop, which SALM5 L327 resides in, likely makes the introduced glycan at this site lack an expected hindrance role as a wedge."

In order for the above statements to be justified by the experimental data, please provide

experimental evidence that these mutations introduce additional N-linked glycosylation sites in the SALM5 mutants (e.g., mass spectrometry). Alternatively, indicate in the text that you do not have experimental evidence that these additional N-linked glycosylation sites are actually glycosylated, and indicate that these mutants may affect the protein in other, different ways, for example, altering the protein fold, and that the latter may provide an explanation for the observed decrease in interaction between SALM5 and PTP δ Ig1-Ig3.

10. Results section p. 14 lines 342 - 343

" His-tagged SALM5 LRR-Ig mutants E365N and K309N/R311T didn't bind to Flag-tagged PTP δ Ig1-3 MeA- displayed on the surface of HEK293T cells,... (Fig. 4c)"

Fig. 4c refers to Myc-tagged PTP δ Ig1-3 MeA-. Please explain or revise.

11. Please indicate in Fig 4a how the SALM5 mutants were produced (cell-line, purification procedure) and if the protein was subsequently treated with Endo H before the pull-down, i.e., what is the source of the SALM5 mutants used for pull-downs? This is not clear from the Methods section.

Please provide an SDS-PAGE (or Western Blot) of the SALM5 mutants used for the pull-downs. The reader should be able to assess whether the amount of SALM5 mutant used is the same for each pull-down and what the molecular weight is of the SALM5 (i.e. the protein is not degraded/fragmented). This is a necessary control and should be part of Fig. 4a.

12. Results section p. 17 lines 357 - 359.

"Unexpectedly, the mutants of SALM5 ectodomain, L327N and E365N, were not expressed in HEK293T cells, as indicated by Western-blotting and immunostaining experiments (Supplementary information, Figure S5)."

Please provide an explanation for why SALM5 L327N and SALM5 E365N don't express in HEK293T cells. How are these proteins different than the proteins used for Fig. 4a?

13. Fig. 2e and Fig. 4d are identical (see top two rows)

Fig. 4d refers to K390N/R311T but instead shows a duplicate of Fig. 2e (mutants R110N/E160S).

14. Results section p. 17 line 360 - 364.

"Compared with the wild-type SALM5, the mutant K309N/R311T significantly decreased intensity of synapsin-I clustering in the hippocampal neuron that HEK293T cells formed artificial synapses with (Figure 4d, 4e).

This reviewer is unable to assess whether this statement is experimentally justified, because Fig. 4d is a duplicate of Fig. 2e, i.e. the relevant data to assess the above statement is not presented in the manuscript.

15. Results section p. 17 line 360 - 364.

"Incidentally, two mutants L327N and E365N didn't lead to synapsin-I clustering (Figure 4f, 4g). Combined together, disrupting the SALM5/PTP δ interaction may impair SALM5-dependent synaptic differentiation."

Given that SALM5 L327N and E365N do not express in HEK293T cells, what is Fig. 4f and 4g meant to demonstrate and what does Fig. 4f and 4g actually show? This should be explained in the text (lines

360-364). The authors should consider moving the data for SALM5 L327N and E365N (Figure 4f, 4g) to the supplement material.

16. Fig. S8 (especially Fig. S8f is very informative) is a beautiful and very insightful figure. This figure could go into the main paper, not in the supplemental material.

17. Given the 3.7 Ang resolution of SALM5 in complex with PTP δ Ig1-Ig3, please removed the distances in Supplemental Table 3 or justify how at this resolution you are able to determine distances between atoms with 0.01 Ang. accuracy.

Minor comments:

1. Please have the manuscript thoroughly proofread. There are many, many grammatical, typographical ("typos"), and stylistic errors.

2. Abstract section p.2 line 29

"human SALM5 LRR-Ig alone and in complex with human PTP δ Ig1-3 domains with the mini-exon B insert."

Introduction section p. 6 line 112

"human PTP δ 112 Ig1-3 (MeA-)"

Please define the PTP δ Ig1-3 protein used in terms of both MeA and MeB and keep consistent throughout paper

3. Results section p. 9 lines 181 – 182

"Different from previous sequence alignment that suggested SALM5 contains six LRRs7, we assumed that SALM5 has seven LRR repeats."

Please refer in the text to Supplemental data Fig. S2a to help the reader.

4. Results section p. 10 lines 207 – 209

" Superimposition of the Ca atoms of free and PTP δ -bound SALM5 highlights a potential dynamic movement of the Ig domain around the linker (Supplementary information, Figure S2e)."

The superposition of Ca atoms does not show this, rather the absence of the majority of the Ig domain in SALM5 when PTP δ -Ig1-Ig3 is not present does suggest this. Please revise the text.

5. Fig. 2a is not easily interpreted.

Please indicate N- and C- termini. Please replace the heavy dotted lines indicating hydrogen bonds with dotted lines that have larger spacing between dots. Please indicate the residues R11, Q134, N158, R23, and Tyr 273 in the figure so that the zoomed-in image in Figure 2b (which is rotated in orientation) can easily be found back in Fig. 2a. It is helpful to colour LRRNT and LRRCT a different colour in Fig. 2a, so that it is clear how they contribute to the dimer interface.

6. Results section p. 14 lines 296 - 297

"Notably, on the edge of Site I, R217 from the splicing inert MeB is in proximity, but not forming salt bridge, with SALM5 E280."

Please indicate the MeB insert in Fig. 3a (label MeB and indicate in the protein structure with a

different coloured segment).

7. Results section p. 15 line 323

“Noteworthy, a double-alanine mutation of S329 and S360 was reported to weaken SALM5 binding¹⁵.”

Please indicate in the text, to which protein partner the mutant S329A/S360A weakened SALM5 binding.

Please also indicate SALM5 S329 and S360 in Fig. 3a and 3d in order to show potential allosteric effect.

Reviewers' comments:

Reviewer #1 (Remarks to the Author):

This study by Lin et al. reports the crystal structure of a 2:2 SALM5/PTPd heterotetramer. The authors find that both the interactions between two LRRs as well as SALM5-PTPd are important for the formation of the tetramer as well as the induction of presynaptic differentiation in contacting axons.

Given the reported association of SALM5 with schizophrenia and autism spectrum disorders, and considering the fact that this is the first crystal structure of the SALM/Lrfrn family proteins that proposes an important hypothesis that induced PTPd dimerization would be important for presynaptic activation of tyrosine phosphatase activity and presynaptic differentiation, these findings are quite interesting and represent a significant step forward in the field.

Major comments:

1. In previous reports (Choi et al, 2016, Scientific Report), the ecto domain of SALM5 strongly binds MeB - forms of LAR-RPTPs (LAR, PTPd, and PTPs). In addition, insertion of MeB dramatically reduces complex formation. The authors solve the structure of PTPd MeB⁺ in complex with SALM5. Since the ligands of LAR-RPTPs showed the splice code specificity, the investigation of splice code selectivity of SALM5 is important to understand and generalize the ligand specificity of LAR-RPTPs by modulating splice code. Hence, the author should provide the binding affinity comparison in terms of splice code variants.

R: Responding to the reviewer's comment, we have added SPR experiments determining the binding affinities of human SALM5 LRR-Ig with human PTPδ Ig1-3 and its variants MeA⁻, MeB⁻ and MeA⁻/MeB⁻. Correspondingly, we have revised the main text discussing the results, added Figure 5 showing the sensorgrams and supplemented experimental description in "Methods".

2. The authors suggest the trans-synaptic interaction between PTPd and SALM5 in the manuscript. However, the SALM5 could be present at both pre- and postsynaptic sites, considering the reported homophilic and intercellular adhesion of SALM5. In addition, Motani et al. (J Neurosci 2017) have recently suggested the possibility that PTPd could be located at postsynaptic sites. Therefore, the possibility of SALM5-PTPd interaction occurring in a cis manner at both pre- and postsynaptic should be discussed.

R: We have revised the "Discussion" and cited the references. Considering the PTPd isoforms mentioned by the reviewer in minor comment #2, we have proposed that the cis-interaction between SALM5 and PTPd (or isoforms) on pre- or post-synaptic neurons might competitively regulate SALM5-mediated synaptic differentiation.

3. In Supp Figure S3b, the intensity of SALM5 R110N/E160S seems not sufficient to figure out the exact elution volume of the peak (blue solid line). Could the author increase the concentration of the mutant and carefully compare with wild type? If the elution volume is similar with that of wild-type protein, the mutant might be a dimer in solution.

R: We have tried many times to upload the SALM5 R110N/E160S samples for gel filtration chromatography at higher concentrations, but failed to recover the proteins with an expected high yield (in a high peak). The underlying reason, we suppose, is that the monomeric SALM5 R110N/E160S would expose some hydrophobic patches, like around L43 and F44, or around L260 and Y273, as shown in Figure 2c, so that its thermodynamic stability is lower than the native dimeric SALM5. As responding to the third reviewer's comments, we have used analytical centrifugation studies to evaluate the relatively precise molecular masses of the native SALM5 and its R110N/E160S mutant.

4. Figure 2d should show similar amounts of WT and mutant SALM5 proteins.

R: We have re-run the SDS-PAGE for similar amounts of WT and mutant SALM5 proteins, and updated this figure.

Minor comments:

1. In Figure 4c,d,e, related text and figure labeling for specific mutations (L317N, E365N, and K309N/R311T) do not match with each other. In addition, the authors are showing in Figure 4d the images that the authors have already shown in Figure 2 again. These apparent mistakes should be corrected.

R: We thank this reviewer for pointing out such a mistake that we carelessly made when we pasted the figures for the original manuscript. Now we have corrected the figures and labeling.

2. The characteristic of the SALM5-PTPd interaction, according to the authors, forces the PTPd molecule to conform a 'V' shape regarding its Ig domains, from which the FNII-like domains then round back to the presynapse. Naturally, the authors assume that for that reach to occur, PTPd will have to display the entire eight FN-like domain chain, as the reach from the presynapse will have to be $\sim 162\text{\AA}$ (Supp Figure 9). This, then, raises questions as to the biological significance of this interaction, because the dominant isoform of PTPd in the brain contains only four FNIII-like domains. This should be considered in interpreting the results and drawing models for Supp Figure 9.

R: We have considered this comment and this reviewer's major comment #2 together in the "discussion".

Firstly, we have added a panel to Supp Figure S9 (now S10) and a sentence to the text saying "Given that the dominant isoform of PTP δ in the developing brain contains only four Fn domains (Yamagata et al., 2015), such a model needs a modification based on flexibility of interdomain linkers in PTP δ and SALM5 (Supplementary Fig. S10b)".

In the supplementary Fig.10b, we have explained how to modify the model, saying “In comparison with the counterpart in the model (a), the PTP δ isoform would adopt a more bent configuration, while the complex core (SALM5 LRR-Ig/PTP δ Ig1-3) would be closer to the presynaptic membrane since SALM5 has an unusual, long flexible linker (~ 50 AA long and rich of serine) connecting its Ig and Fn domains.”

Secondly, we have pointed out the biological significance of this model lies in its application to the whole LAR-RPTP family. Since the key residues in PTP δ for SALM5 binding are highly conserved among the LAR-RPTP family that may share a similar topology, this model will also be a template for understanding interactions between SALM5 and other LAR-RPTP family members. Universality of such a model may explain the recent report that a double-alanine mutation of S329 and S360 not only impaired the interaction of SALM5 with all the three LAR-RPTP family members, but also weakened SALM5-mediated synaptic differentiation that is governed by the whole LAR-RPTP family (Choi et al., 2016).

Thirdly, given that some isoforms of PTPd contain only one Ig domain (the related reference, Mizuno et al., 1994 has been added), that SALM5 may engage in intercellular adhesion (Seabold et al., 2008), and that PTPd could be located at postsynaptic sites (mentioned by this reviewer in major comment #2), we have proposed that, similar to cis-PTP δ /IL1RAPL1 interaction antagonizing IL1RAPL1 function (Montani et al., 2017), cis-interaction between SALM5 and PTPd (or isoforms) on pre- or post-synaptic neurons might competitively regulate SALM5-mediated synaptic differentiation.

3. In addition to Kwon et al, the Woo et al paper (Nat Neurosci, 2009) should also be cited.

R: We have added this reference to the text (numbering 24).

4. Page3, line 55; the author should provide the full name when the term appears for the first time in the manuscript; CNS, Central Nerve System.

R: We agree with this reviewer’s suggestion and have corrected it.

5. Line 177. I wonder the "LRRNT" should be "LRRCT".

R: Yes, this reviewer is right. We have corrected it.

6. Line 251. Typo. Monomeric.

R: We have corrected it.

7. Line378. SALM5 family should be changed to SALM family.

R: We have corrected it.

Reviewer #2 (Remarks to the Author):

This paper, by Lin et al, describes the first crystal structures of the ectodomain region of a SALM-family protein, SALM5, both in isolation and in complex with the

LAR-family receptor tyrosine phosphatase PTP δ . SALM5 is thought to impact neuronal differentiation by interacting with LAR-RPTPs, and the current paper begins to reveal the corresponding mechanism in atomic detail. The 2.8Å and 3.7Å structures of SALM5 and its complex with PTP δ appear to be of high quality (although free R factors appear somewhat high). The complex structure shows that SALM5 is a strongly associating rod-shaped dimer, and the rod-shaped SALM5 dimer is bound at either end by a single PTP δ molecule, thus “dimerizing” PTP δ , with an intervening space of about 40Å (the length of the SALM5 rod). The paper validates this interpretation with site-directed mutagenesis and solution biophysical measurements, and conservation suggests other SALM-family members may function similarly. Functional experiments also confirm the model, showing that formation of the crystallographically observed complex is necessary to induce the recruitment of synapsin.

Overall, this is an excellent study on an interesting subject, which I think to be appropriate for publication in Nature Communications.

Reviewer #3 (Remarks to the Author):

Manuscript: # 133492

Authors: Zhaohan Lin, Jianmei Liu, Huandi Ding, Fei Xu, Heli Liu

Title of manuscript: Structural basis of SALM5-induced PTP δ dimerization for synaptic differentiation

This manuscript presents the crystal structures of human SALM5 LRR-Ig alone and in complex with human PTP δ Ig1-3 domains with the mini-exon B insert. SALM5 and PTP δ are two synaptic adhesion molecules that play a role in synapse development, and are associated with neuropsychiatric diseases.

To substantiate the structural findings, the authors carry out biochemical experiments including structure-guided mutagenesis and heterologous synapse formation assays are carried out.

The key findings that this manuscript would like to present are the following:

- the structures of SALM5 (2.8 Å) and its complex with the fragment human PTP δ Ig1-3 (3.7 Å)
- while the Ig domain in SALM5 is likely flexible, in the complex with PTP δ Ig1-3, the Ig2 and Ig3 from PTP δ wrap around the SALM5 Ig domain fixing the latter's conformation.
- SALM5 exists as a dimer in solution, mediated by the LRR domains which pack in an antiparallel fashion
- dimerization of SALM5 is prerequisite for its functionality in inducing synaptic differentiation.
- SALM5 directly induces cis-dimerization of LAR-RPTPs into higher-order

signaling assembly which mediate the clustering of synapsin I (as a readout for presynaptic differentiation).

Significance: The mechanism of SALM5 function and its interaction with PTP δ has important implications for how these synaptic adhesion molecules can promote the formation of synapses.

The structural work is beautiful. Concern, however, exists that important controls in the biochemical and cell-based assays are not included in the manuscript (as discussed below) which makes it difficult to use these experimental data to solidly support the (structural and functional) conclusions from the manuscript that 1) SALM5 is dimeric in solution, 2) the interaction between SALM5-PTP δ Ig1-Ig3 as seen in the crystal structure represents interactions in the physiological complex, and 3) if you disrupt SALM5-PTP δ Ig1-Ig3 interaction, you prevent synapsin I clustering (as a read-out for presynaptic differentiation).

R: Here we would like to generally respond to this reviewer's major concerns.

First, as cited in the "introduction", Zhu et al has demonstrated that SALM5 is a dimer in solution (Zhu et al., 2016). This is consistent with that SALM5 was a dimer in crystals. Combined with the gel-filtration data in our original manuscript, the updated chemical cross-linking reaction and the supplemented analytical ultracentrifugation data in this revised version (described below) further confirmed that SALM5 mainly exists as a dimer in solution.

Second, our crystal structures of SALM5 alone and in complex with PTP δ revealed the consistent dimerization mode of SALM5 in different crystals. Structure-based introduction of "glycan wedges" to the dimer interface successfully dissociated the dimer, as convinced by cross-linking reaction, analytical ultracentrifugation and mass spectrometry data in this revised manuscript. Surface plasma resonance, protein-cell interaction and heterologous synapse formation assays have showed that structure-based mutagenesis to disrupt SALM5 dimerization or SALM5-PTP δ interaction may impair SALM5-mediated synaptic differentiation. From this point, we can make a conclusion that dimerization of SALM5 and SALM5/PTP δ interaction as seen in the crystal structures may present a state in physiological context.

Third, we frankly admit that the disrupted interaction, which we tested using SPR, pull-down assay and cell surface binding assays, is just between SALM5 and PTP δ , and that the interactions between SALM5 and other LAR-RPTP family members would also regulate SALM5-mediated presynaptic differentiation (synapsin I clustering as a read-out). However, because the key residues in PTP δ for SALM5 binding are highly conserved among the LAR-RPTP family that may share a similar topology, we propose that the SALM5/PTP δ interaction model (Supplementary information, Figure S10) would also be suitable for understanding interactions between SALM5 and other LAR-RPTP family members. Universality of such a model may explain the recent report that a double-alanine mutation of S329 and S360 not only impaired the interaction of SALM5 with all the three LAR-RPTP family members, but also decreased SALM5-mediated synaptic differentiation (Choi et al., 2016).

Correspondingly, we have revised the manuscript and responded to this reviewer's comments as follows.

Major comments:

1. Results section p. 8 lines 155-157

“... PTP δ -bound SALM5 structures unanimously revealed that two SALM5 molecules form a diad dimer through their LRR domains packing in an antiparallel side-by-side manner.”

The supplemental figure S1c and S1d does not suffice as evidence for the statement above, because the reader cannot easily see how different or similar the SALM5 dimer is alone versus in the complex with PTP δ Ig1-Ig3.

Please provide a figure whereby the SALM5 dimer (from the structure of the SALM5+PTP δ Ig1-Ig3 complex) is superimposed on top of the SALM5 dimer (from the structure of SALM5 alone).

This is crucial because the SALM5 dimer in the SALM5+PTP δ Ig1-Ig3 complex is apparently generated by non-crystallographic symmetry, however, the SALM5 dimer in the structure of SALM5 alone is generated by crystallographic symmetric.

R: Following this reviewer's suggestion, we have added a panel (labeled as “e”) in Supplementary Figure S1 to show superimposition of a non-crystallographic 2-fold axis related SALM5 dimer (bound with PTP δ) with a crystallographic 2-fold axis related SALM5 dimer (alone). The structures of the two dimers can be aligned well with an RMSD of 0.69 Å for 266 paired Ca atoms.

2. Results section p. 8 lines 157 – 161.

“The LRR domains of two SALM5 molecules constitute a concave platform as the core of the dimer, with approximate dimensions of 56 Å \times 68Å. In each SALM5 protomer, the Ig domain keeps its long axis perpendicular to the platform; the two Ig domains in the SALM5 dimer are in parallel with each other and no contacts are observed between them (Figure 1d).”

Please revise the text to make clear that you are referring to SALM5 as seen in the complex with PTP δ Ig1-Ig3. This is important because in absence of PTP δ Ig1-Ig3, the Ig domain in SALM5 is not visible.

R: Following the reviewer's advice, we have made it clear in the text through adding the phrase “PTP δ -bound”.

3. In the legend of Fig. 1d, please indicate in the text that you are showing SALM5 as it is seen in the context of the complex with PTP δ Ig1-Ig3.

For each figure legend throughout the manuscript indicate whether you are showing the structure of SALM5 alone (2.8 Ang. structure) or SALM5 extracted from the complex (3.7 Ang. structure).

R: Taking the reviewer's suggestion, we have distinguished structure source using the phrases “extracted from PTP δ -bound complex structure” or “from the 2.8 Å free SALM5 structure”.

4. Results section p. 12 line 249

“Since SALM5 mainly exists in solution as a dimer,....”

Please show experimental evidence for this statement, for example, running the SALM5 sample on a size exclusion column in parallel with gel filtration calibration protein standards or analytical centrifugation studies.

Comparison of Fig. S3a and S3b does not suffice, because calibration markers for the chromatographic runs in Fig. S3 are lacking, so it is not possible to correlate elution volume with estimated molecular weight. Calibration markers are also needed to be able to interpret if a difference between 13.8 ml and 14.3 ml is significant between wild type SALM5 and SALM5 R110N/E160S (Fig. S3a and S3b), suggesting dimeric versus monomeric states respectively.

R: Firstly, we have cited the recently paper (Zhu et al., 2016) in the text to show that SALM5 has been reported to exist as a dimer in solution, in line with our two crystal structures.

Secondly, we have used analytical centrifugation study to precisely measure the molecular masses of the wild-type human SALM5 LRR-Ig and its variant R110N/E160S. The results indicated that the R110N/E160S mutant is monomeric.

Thirdly, as responding to the first reviewer’s comment #3, likely due to exposure of hydrophobic patch to solvent, the R110N/E160S mutant had low recovery yield from gel filtration chromatography. Therefore, we didn’t compare the molecular masses of the wild-type SALM5 and the R110N/E160S variant using gel filtration calibration markers.

5. SALM5 R110N/E160S (Fig. 2d) does not appear more heavily glycosylated, which would be expected, if the glycan wedge was formed in order to disrupt dimer formation of SALM5.

Please demonstrate that R110N/E160S are N-linked glycosylated or explain in the text how the exact protein samples run on the gel filtration columns in Fig. S3b were expressed and purified. Please explain in the text if the proteins used for Fig. S3 were expressed in HEK293 GnTI- cells and EndoH treated (i.e. lacking glycosylation)?

Can you rule out that R110N/E160S cause folding defects?

The markers on the SDS-PAGE gel inserts for Fig. S3a and S3b seem different. Is this correct?

R: First, as responding to the first reviewer’s major comment #4, we have re-run the SDS-PAGE for similar amounts of WT and mutant SALM5 proteins, and updated Figure 2d. This figure clearly showed that SALM5 R110N/E160S appeared more heavily glycosylated since it migrated more slowly than the wild type on the gel. As shown in Figure S3a, N-linked glycan introduced at N110 was convinced by mass spectrometry.

In the “Cloning, baculovirus recombination and protein preparation” section in “Methods”, we have annotated that “except for crystallization, all the purified recombinant proteins were not applied to glycan-trimming”, and that “Proteins used for crystallization, gel filtration assay, mass spectrometry, analytical centrifugation study, pull-down assay, surface plasma resonance and cell-binding assays were

expressed...". These annotations will more clearly show how the proteins used for gel filtration analysis in this figure were produced.

Second, we can rule out that R110N/E160S didn't cause folding defects because it remains function of binding to PTP δ as shown in gel-filtration chromatography.

Third, yes, the markers were different.

6. Because the cross-linked SALM5 is a minor band in Fig. 2d, it is important to load the same amount of protein for the mutant SALM5 R110N/E160S. Approximately 4x more wild type SALM5 is loaded on the gel compared to the mutant SALM5 R110N/E160S, so a cross-linked species might not been seen because insufficient sample is loaded for the mutant SALM5 R110N/E160S.

Please justify or re-run the SDS-PAGE gel.

R: As responding to the above comment #5, we have re-run the SDS-PAGE gel and updated the panel Figure 2d.

7. Results section p. 12 line 259

"The monomeric SALM5 LRR-Ig is still capable of binding to PTP δ ,..."

Unless the text can be revised to accommodate points 4, 5 and 6 above, the text should be corrected to read "SALM5 R110N/E160S, which we believe is possibly monomeric, is still capable of binding to PTP δ ,..."

R: Adding or updating mass spectrometry, analytical centrifugation and cross-linking reaction assays, we think that we have accommodated the above points and believe that SALM5 R110N/E160S is monomeric in solution.

8. Fig. 2e. Please display a brightfield picture showing the image of the HEK293T cell and the neuron that are co-cultured and being imaged. This is important in order to demonstrate that synapsin I is being clustered on a neuron. In the current image, it is not clear where the neuron boundary is.

Please also indicate in the legend of Fig. 2e which proteins are HA-tagged.

R: As seen in Figure 4f, we have labeled the location of neurons in the brightfield that was judged by synapsin I-staining, and added annotation in the legend, saying "In the bright field, the locations of primary neurons are indicated using yellow arrows." For clarity, in the revised manuscript, we have not provided a brightfield image for Figure 2e that saw a similar distribution of neurons.

In addition, we have indicated in the legend of Fig.2d that SALM5 and its mutant are HA-tagged.

9. Results section p. 14 lines 332 – 333, line 345 and lines 349-353.

" we introduced N-linked glycan wedges to the interface through mutating PTP δ -interacting residues of SALM5 (K309N/R311T L327N and E365N)"

" In summary, disrupting interactions involved by SALM5 K309, R311 and E365 by introducing a glycan to these sites, ..."

"In contrast, introduction of a glycan to SALM5 L327 almost didn't alter PTP δ -binding capacity of SALM5. This meant that hydrophobic interactions involved

by SALM5 L327 are not critical to SALM5/PTP δ recognition. Meanwhile, the intrinsic flexibility of the Ig CD loop, which SALM5 L327 resides in, likely makes the introduced glycan at this site lack an expected hindrance role as a wedge.”

In order for the above statements to be justified by the experimental data, please provide experimental evidence that these mutations introduce additional N-linked glycosylation sites in the SALM5 mutants (e.g., mass spectrometry). Alternatively, indicate in the text that you do not have experimental evidence that these additional N-linked glycosylation sites are actually glycosylated, and indicate that these mutants may affect the protein in other, different ways, for example, altering the protein fold, and that the latter may provide an explanation for the observed decrease in interaction between SALM5 and PTP δ Ig1-Ig3.

R: we have used mass spectrometry to convince that SALM5 mutants, K309N/R311T L327N and E365N are N-linked glycosylated as expected. Please check the MS/MS spectra in Supplementary Figure S3. Correspondingly, we have added “The presence of the glycan wedge, GlcNAc2Man5, for each mutant was confirmed using mass spectrometry (Supplementary information, Figure S3)” in the text.

10. Results section p. 14 lines 342 - 343

“ His-tagged SALM5 LRR-Ig mutants E365N and K309N/R311T didn’t bind to Flag-tagged PTP δ Ig1-3 MeA- displayed on the surface of HEK293T cells,... (Fig. 4c)”

Fig. 4c refers to Myc-tagged PTP δ Ig1-3 MeA-. Please explain or revise.

R: We have revised the text to “Myc-tagged”.

11. Please indicate in Fig 4a how the SALM5 mutants were produced (cell-line, purification procedure) and if the protein was subsequently treated with Endo H before the pull-down, i.e., what is the source of the SALM5 mutants used for pull-downs? This is not clear from the Methods section.

Please provide an SDS-PAGE (or Western Blot) of the SALM5 mutants used for the pull-downs. The reader should be able to assess whether the amount of SALM5 mutant used is the same for each pull-down and what the molecular weight is of the SALM5 (i.e. the protein is not degraded/fragmented).

This is a necessary control and should be part of Fig. 4a.

R: In the “Methods” section of the original manuscript, we have written that “Proteins used for crystallization, gel filtration assay, pull-down assay and surface plasma resonance experiments were expressed in the BacMam system”. According to the context, these recombinant proteins were expressed in HEK293S GnTI cells as we have also described. Now we have updated the first sentence in this section since we added new methods like mass spectrometry and analytical centrifugation. Meanwhile, as responding to this reviewer’s comments #5, we have added a sentence in this section, saying “All the recombinant proteins were purified sequentially using Ni²⁺-NTA affinity and gel-filtration chromatography. Except that used for crystallization, all the purified recombinant proteins were not applied to

glycan-trimming”.

In addition, we have added a panel in Figure 4a to show the SDS-PAGE bands of SALM5 wild-type and its mutants, as well as PTP δ -Fc.

12. Results section p. 17 lines 357 – 359.

“Unexpectedly, the mutants of SALM5 ectodomain, L327N and E365N, were not expressed in HEK293T cells, as indicated by Western-blotting and immunostaining experiments (Supplementary information, Figure S5).”

Please provide an explanation for why SALM5 L327N and SALM5 E365N don't express in HEK293T cells. How are these proteins different than the proteins used for Fig. 4a? degradation occurs to the full-length ectodomain?

R: Different from the proteins used for Figure 4a that include LRR and Ig domains of SALM5, the SALM5 L327N and SALM5 E365N in Figure S5 refer to mutants of the full-length ectodomain of SALM5. As guessed by this reviewer, we did observe the degradation of the full-length SALM5 ectodomain when we purified it for crystallization. Correspondingly, we have added the following sentence to the supplementary Figure S7:

“Due to degradation that occurred to the recombinant full-length SALM5 ectodomain, L327N and E365N mutations would likely accelerate such a degradation so that the mutants couldn't be tested out on the transfected HEK293T cells”.

13. Fig. 2e and Fig. 4d are identical (see top two rows)

Fig. 4d refers to K390N/R311T but instead shows a duplicate of Fig. 2e (mutants R110N/E160S).

R: As corresponding to this reviewer's comment #14 and the first reviewer's minor comment #1, we have replaced this panel with the right one.

14. Results section p. 17 line 360 – 364.

“Compared with the wild-type SALM5, the mutant K309N/R311T significantly decreased intensity of synapsin-I clustering in the hippocampal neuron that HEK293T cells formed artificial synapses with (Figure 4d, 4e).

This reviewer is unable to assess whether this statement is experimentally justified, because Fig. 4d is a duplicate of Fig. 2e, i.e. the relevant data to assess the above statement is not presented in the manuscript.

R: As corresponding to this reviewer's comment #13 and the first reviewer's minor comment #1, we have replaced this panel with the right one.

15. Results section p. 17 line 360 – 364.

“Incidentally, two mutants L327N and E365N didn't lead to synapsin-I clustering (Figure 4f, 4g). Combined together, disrupting the SALM5/PTP δ interaction may impair SALM5-dependent synaptic differentiation.”

Given that SALM5 L327N and E365N do not express in HEK293T cells, what is Fig. 4f and 4g meant to demonstrate and what does Fig. 4f and 4g actually show? This should be explained in the text (lines 360-364). The authors should consider moving

the data for SALM5 L327N and E365N (Figure 4f, 4g) to the supplement material.
R: Here we would like to keep Figure 4f and 4g in the main text because (1) the two panels show that the mutants L327N and E365N truly didn't lead to synapsin-I clustering in the cell-based assay; (2) as corresponding to this reviewer's comment #8, the panels may show the distribution of neurons in the brightfield.

16. Fig. S8 (especially Fig. S8f is very informative) is a beautiful and very insightful figure. This figure could go into the main paper, not in the supplemental material.
R: We agree with this reviewer's suggestion and have moved FigS8 into the main paper as Figure 6.

17. Given the 3.7 Ang resolution of SALM5 in complex with PTP δ Ig1-Ig3, please removed the distances in Supplemental Table 3 or justify how at this resolution you are able to determine distances between atoms with 0.01 Ang. accuracy.
R: We agree with this reviewer's point and have rounded the distance values in this table with 0.1-angstrom accuracy.

Minor comments:

1. Please have the manuscript thoroughly proofread. There are many, many grammatical, typographical (“typos”), and stylistic errors.

R: we have followed this reviewer's suggestion and tried our best to make the manuscript more comfortable to read.

2. Abstract section p.2 line 29

“human SALM5 LRR-Ig alone and in complex with human PTP δ Ig1-3 domains with the mini-exon B insert.”

Introduction section p. 6 line 112

“human PTP δ 112 Ig1-3 (MeA-)”

Please define the PTP δ Ig1-3 protein used in terms of both MeA and MeB and keep consistent throughout paper

R: Following this reviewer's suggestion, we have defined PTP δ Ig1-3 (MeA) as PTP δ Ig1-3 domain with MeA splice insert deleted.

3. Results section p. 9 lines 181 – 182

“Different from previous sequence alignment that suggested SALM5 contains six LRRs7, we assumed that SALM5 has seven LRR repeats.”

Please refer in the text to Supplemental data Fig. S2a to help the reader.

R: Following this reviewer's suggestion, we have referred to Figure 1e and Supplemental data Fig. S2 which show 7 LRR repeats and how these repeats are separate from LRRNT and LRRCT.

4. Results section p. 10 lines 207 – 209

“ Superimposition of the Ca atoms of free and PTP δ -bound SALM5 highlights a

potential dynamic movement of the Ig domain around the linker (Supplementary information, Figure S2e).”

The superposition of Ca atoms does not show this, rather the absence of the majority of the Ig domain in SALM5 when PTP δ -Ig1-Ig3 is not present does suggest this. Please revise the text.

R: In the original Figure S2e, we have used an arrow to indicate such a potential domain movement, and taking this reviewer’s suggestion, we have revised the text by adding “which is further suggested by the absence of the majority of the Ig domain in free SALM5 structure”.

5. Fig. 2a is not easily interpreted.

Please indicate N- and C- termini. Please replace the heavy dotted lines indicating hydrogen bonds with dotted lines that have larger spacing between dots. Please indicate the residues R110, Q134, N158, R23, and Tyr 273 in the figure so that the zoomed-in image in Figure 2b (which is rotated in orientation) can easily be found back in Fig. 2a. It is helpful to colour LRRNT and LRRCT a different colour in Fig. 2a, so that it is clear how they contribute to the dimer interface.

...label residues R110 and N158 to which glycans were introduced through mutation.

R: Following the reviewer’s suggestions, we have revised Figure 2a by indicating N- and C-termini, updating the dotted lines and changing the colors of LRRNT and LRRCT. Since Figure 2a is already a little bit noisy, we have just labeled R110 and N158 from one SALM5 molecule for clarity.

6. Results section p. 14 lines 296 - 297

“Notably, on the edge of Site I, R217 from the splicing inert MeB is in proximity, but not forming salt bridge, with SALM5 E280.”

Please indicate the MeB insert in Fig. 3a (label MeB and indicate in the protein structure with a different coloured segment).

R: For Fig.3a and Fig.3b, we have colored the MeB insert in slate.

7. Results section p. 15 line 323

“Noteworthy, a double-alanine mutation of S329 and S360 was reported to weaken SALM5 binding¹⁵.”

Please indicate in the text, to which protein partner the mutant S329A/S360A weakened SALM5 binding.

Please also indicate SALM5 S329 and S360 in Fig. 3a and 3d in order to show potential allosteric effect.

R: (1) We have indicated in the text that the mutant S329A/S360A weakened SALM5 binding to LAR, PTP δ and PTP σ . (2) We have added supplementary Figure S6c-d to show the location of S329 and S360 as in Fig.3a and 3d.

REVIEWERS' COMMENTS:

Reviewer #1 (Remarks to the Author):

The authors have successfully addressed all of my comments. I appreciate their efforts for the revision, but I have the following small final comments.

1. Line 287 says "the MeB insert was reported to strongly inhibit SALM5/PTPd interaction", but this is misleading. The actual data in the paper show that the interactions are merely reduced, not strongly reduced.
2. A possible explanation for the roles of MeA/B splice variant in the SALM5-PTPd interaction could be the different methods employed to assess the interaction; i.e. L-cell aggregation vs. purified proteins. This should be discussed.
3. The labelings for MeA/B in Figure 5 are confusing. It would be nice if the authors could indicate the splice variants in the following way; i.e. "MeA-" can be changed to "MeA-B+".

Reviewer #3 (Remarks to the Author):

The authors have done a good job responding to the reviewer comments.

Point #4:

In response to reviewers, the authors write: "Secondly, we have used analytical centrifugation study to precisely measure the molecular masses of the wild-type human SALM5 LRR-Ig and its variant R110N/E160S."

Please note that analytical ultracentrifugation sedimentation velocity experiments do not precisely tell you the molecular masses of proteins, because as the authors note, the shape of the molecules influences the S-value used to extract an 'apparent molecular weight'. You need analytical ultracentrifugation sedimentation equilibrium experiments to precisely measure the molecular masses of the proteins.

The authors would be very wise to indicate in the text that they realize this shortcoming, and that their sedimentation velocity studies are highly suggestive, but do not prove their statement that wild type SALM5 is a dimer, while the mutant R110N/E160S is a monomer.

Please fix through out text:

'as convinced by mass spectrometry'

should read

'as demonstrated by mass spectrometry'

REVIEWERS' COMMENTS:

Reviewer #1 (Remarks to the Author):

The authors have successfully addressed all of my comments. I appreciate their efforts for the revision, but I have the following small final comments.

1. Line 287 says "the MeB insert was reported to strongly inhibit SALM5/PTPd interaction", but this is misleading. The actual data in the paper show that the interactions are merely reduced, not strongly reduced.

R: We have taken this reviewer's comment and omitted the word "strongly" in the text.

2. A possible explanation for the roles of MeA/B splice variant in the SALM5-PTPd interaction could be the different methods employed to assess the interaction; i.e. L-cell aggregation vs. purified proteins. This should be discussed.

R: Responding to the reviewer's requirement, we have added the sentence saying "Different from the previous L-cell aggregation assay-based report that meB in PTPδ additively suppressed PTPδ binding to SALM5, our SPR experiments revealed that the addition of meB to PTPδ significantly enhanced its binding to SALM5. This difference might derive from the different methods employed to assess SALM5/PTPd interaction." Since there have been 4992 words already in the main text, the above sentence has been added to the legend of Figure 5 due to word limitation (no more than 5000 words for the main text).

3. The labelings for MeA/B in Figure 5 are confusing. It would be nice if the authors could indicate the splice variants in the following way; i.e. "MeA-" can be changed to "MeA-B+".

R: Adopting this reviewer's suggestion, we have changed the labelings for MeA/B in the panels (a), (b) and (c) of Figure 5 into "MeA⁺/MeB⁺", "MeA⁻/MeB⁺" and "MeA⁺/MeB⁻", respectively.

Reviewer #3 (Remarks to the Author):

The authors have done a good job responding to the reviewer comments.

Point #4:

In response to reviewers, the authors write: "Secondly, we have used analytical centrifugation study to precisely measure the molecular masses of the wild-type human SALM5 LRR-Ig and its variant R110N/E160S."

Please note that analytical ultracentrifugation sedimentation velocity experiments do

not precisely tell you the molecular masses of proteins, because as the authors note, the shape of the molecules influences the S-value used to extract an 'apparent molecular weight'. You need analytical ultracentrifugation sedimentation equilibrium experiments to precisely measure the molecular masses of the proteins.

The authors would be very wise to indicate in the text that they realize this shortcoming, and that their sedimentation velocity studies are highly suggestive, but do not prove their statement that wild type SALM5 is a dimer, while the mutant R110N/E160S is a monomer.

R: Thanks to this reviewer's reminding, we have added the sentence to the legend of the Supplementary Figure 4, saying "A caveat should be given here that sedimentation velocity experiment has intrinsic limitations in deducing molecular masses of proteins. However, since there is one dominant species in both SALM5 WT and mutant sample solutions, the $c(M)$ distribution can be derived from the $c(s)$ distribution for molecular mass calculation." Correspondingly, we have also added the phrase to the legend "this sedimentation velocity study suggests that".

Please fix throughout text: 'as convinced by mass spectrometry' should read 'as demonstrated by mass spectrometry'

R: We have changed the word "convinced" to "demonstrated".